# Magnetic cilia carpets with programmable metachronal waves

Hongri Gu [1], Quentin Boehler [1], Haoyang Cui[1], Eleonora Secchi[2], Giovanni Savorana[2], Carmela De Marco[1], Simone Gervasoni[1], Quentin Peyron[3,4], Tian-Yun Huang [1], Salvador Pane [1], Ann M. Hirt [5], Daniel Ahmed[1] & Bradley J. Nelson[1✉]

Metachronal waves commonly exist in natural cilia carpets. These emergent phenomena, which originate from phase differences between neighbouring self-beating cilia, are essential for biological transport processes including locomotion, liquid pumping, feeding, and cell delivery. However, studies of such complex active systems are limited, particularly from the experimental side. Here we report magnetically actuated, soft, artificial cilia carpets. By stretching and folding onto curved templates, programmable magnetization patterns can be encoded into artificial cilia carpets, which exhibit metachronal waves in dynamic magnetic fields. We have tested both the transport capabilities in a fluid environment and the loco-motion capabilities on a solid surface. This robotic system provides a highly customizable experimental platform that not only assists in understanding fundamental rules of natural cilia carpets, but also paves a path to cilia-inspired soft robots for future biomedical applications.

[1] Institute of Robotics and Intelligent System, ETH Zurich, 8092 Zurich, Switzerland. [2] Institute of Environmental Engineering, ETH Zurich, 8093 Zurich, Switzerland. [3] ICube Lab, UDS-CNRS-INSA, 67400 Illkirch-Graffenstaden, France. [4] FEMTO-ST Institute, Université Bourgogne, Franche Comte, CNRS, 25000 Besançon, France. [5] Institute of Geophysics, ETH Zurich, 8092 Zurich, Switzerland. ✉email: bnelson@ethz.ch

Metachronal waves are self-organized rhythmic patterns appearing in systems with large numbers of hair-like structures, typically found in motile cilia on sea animals and microorganisms[1,2], hairs on epidermal surfaces[3], and the legs of walking millipedes[4]. For motile cilia, this non-reciprocal motion can collectively move fluid in their vicinity, promoting faster and more efficient pumping compared to identically beating cilia arrays[5]. Fluidic transport at the microscale, such as this, is critical for biological processes, including mucus clearance in the human respiration system[6–9], egg transport in fallopian tubes[10], mass transport in coral reefs[3], and microorganism swimming[2,11].

Recent work reveals that biosystems organize different cilia beating patterns to realize more complex functions. For example, the Hawaiian bobtail squid uses two cilia beating patterns to actively recruit their symbiotic bacteria *Vibrio fischeri*[12]. Motile cilia first beat in metachronal waves to create large fluidic vortices to focus micrometre-sized particles into the sheltered zone. After this, randomly beating cilia in the sheltered zone promote colonization of *Vibrio fischeri* by efficiently mixing chemical signals[13]. Bobtail squids recruit this bioluminescent bacteria in order to glow at night and confuse predators as moonlight[12]. In another example, starfish larvae can dynamically change their cilia beating patterns to balance between swimming and food capture[14]. The cilia band can be switched to beat in opposite directions, controlling vortex arrays around the animal, which provides a unique advantage in complex environments.

To date, these emergent phenomena of cilia carpets, including metachronal waves, are primarily studied through numerical simulations[5,15–18] and observations of natural ciliated organisms[3,13,14,19]. Natural organisms provide a limited selection of cilia hair properties (e.g. hair length, density), making it difficult to investigate customized cilia interactions. Natural cilia systems are usually sensitive to the strong light used in high-speed imaging. Alternatively, artificial cilia provide ways to engineer customized cilia systems. Over the past 20 years, multiple artificial cilia systems have been developed, driven by pneumatic[20], light[21], acoustics[22], electric fields[23], and magnetic fields[24,25]. Many of them have found microfluidic applications as micropumps and micromixers[26–28]. For magnetic artificial cilia, Hanasoge et al. built a line of magnetic cilia with different lengths, whose distinct bending thresholds give rise to metachronal motion in a uniform magnetic field[29]. Tsumori et al. demonstrated metachronal waves in two-dimensions by an assembly of pre-fabricated cilia hairs[30].

Most artificial cilia carpets are limited to one of the two categories: systems with a large number of cilia beating in phase[31–33] or with a limited cilia number and a few customizable parameters[34–36]. To study complex emergent phenomena, especially metachronal waves, both large numbers and programmable cilia moving patterns are necessary. Here we present a soft robotic approach for cilia research. First, we present a simple and scalable method to fabricate stretchable magnetic cilia carpet with a large number (>200) of cilia hair structures. The stretchable cilia carpet is composed of hairs made of a magnetic composite material (NdFeB and Ecoflex) and a non-magnetic stretchable substrate (pure Ecoflex). By stretching the carpet to conform to various three-dimensional (3D) geometries, we can encode complex magnetization patterns in the cilia arrays using a magnetizer. The complex magnetization profile on the cilia carpet will later translate to metachronal wave patterns under a dynamic magnetic field. Unlike metachronal waves that are self-organized in natural cilia, metachronal waves on magnetic cilia carpets are only determined by their magnetization patterns. We characterize the magnetic cilia motions under dynamic magnetic fields and build a model to predict the metachronal motions for various complex magnetization patterns. To show the potential of this soft robotic system in cilia research, we have tested both the transport capabilities in a fluid environment and the locomotion capabilities on a solid surface. The experimental study of liquid transport revealed a dependence of the propulsion velocity on metachronal wavelength and on cilia density, as reported in previous numerical studies[5,17]. The surface propulsion of our soft robot shows a locomotion mechanism similar to that of the giant African millipede. This work provides an experimental platform for customized cilia carpets with a large design space to facilitate fundamental studies of natural cilia, cilia-based active matter systems, and cilia-inspired soft robotic applications.

## Results

**Stretchable soft magnetic cilia carpets.** The magnetic cilia hair has a cylindrical shape with a diameter of 0.8 mm and a length of 4 mm. To fabricate soft cilia carpets, we used a two-step moulding on 3D printed polymer structure (Fig. 1a). After printing, the mould was cleaned to remove all supporting material and then coated with a thin layer of resin to smooth the surface (Fig. 1b). After the resin was cured, the mould was cooled to room temperature. In the first moulding step, NdFeB particles and Ecoflex 00-30 were mixed in a 1:1 weight ratio. The thick mixture was degassed in a vacuum chamber and filled into the mould. The mixture was trapped in the small diameter (0.8 mm) cylindrical holes, and excess mixture was removed. The second moulding step fills a carpet substrate volume with pure Ecoflex 00–30, making the cilia substrate soft and highly stretchable. The mould was then placed in an oven to fully cure the Ecoflex at the glass transition temperature of the mould material[37]. After curing and cooling to room temperature, the cilia carpet can be easily peeled off the mould without damaging the magnetic hairs.

The soft cilia carpets contain non-magnetized NdFeB particles (average diameter 5 μm). These particles are not pre-magnetized so the cilia will not move under the actuating magnetic field (80 mT). To encode magnetization information on the cilia carpet, we magnetized the carpet by using a template with predefined curvature[38]. Each sample was placed in the magnetizer, and the maximum pulse magnetic field (1.2 T) was applied.

Compared with previously reported magnetic cilia[24,29,30,38], our system has distinct advantages: First, we show a simple two-step silicone moulding that is repeatable and capable of fabricating large numbers of cilia at the millimetre scale. Second, there is no assembly process required in the fabrication process. Furthermore, our method provides customizable parameters for individual cilium due to the use of 3D printing technology. To demonstrate this capability, we fabricate cilia carpets with densely packed hairs (Fig. 2a right) and cilia carpet with different hair lengths across the carpet (Fig. 2a left) to show the flexibility in the design space. Finally, the highly stretchable cilia substrate (Fig. 2b) allows the carpet to conform to various template shapes to encode complex magnetization patterns in the magnetizer (Figs. 1 and 3, Supplementary Fig. 6).

**Magnetization and motion of single cilium.** In this study, we use a slowly rotating magnetic field (30 degrees/s) to drive the artificial cilia. The motion of artificial cilia exhibits two phases within one cycle, similar to the "power stroke" and "recovery stroke" in natural cilia[39]. The two phases consist of a synchronized phase and an asynchronized phase. In the synchronized phase, the cilia follow the external magnetic field moving from one side to the other. During the asynchronized phase, the cilia cannot continue following the magnetic field due to boundary constraints and "snap back" to their original position.

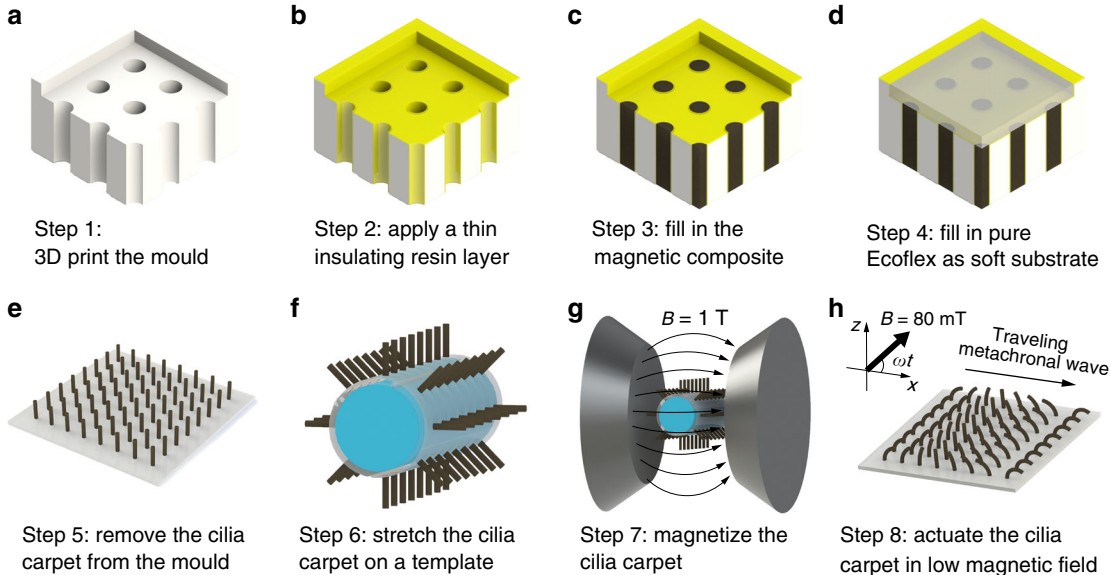

**a** Step 1: 3D print the mould

**b** Step 2: apply a thin insulating resin layer

**c** Step 3: fill in the magnetic composite

**d** Step 4: fill in pure Ecoflex as soft substrate

**e** Step 5: remove the cilia carpet from the mould

**f** Step 6: stretch the cilia carpet on a template

**g** Step 7: magnetize the cilia carpet

**h** Step 8: actuate the cilia carpet in low magnetic field

**Fig. 1 Schematic process of fabrication and magnetization of artificial magnetic cilia carpets. a**, **b** The mould is 3D printed and coated with a thin layer of insulating resin. The insulating layer smooths the surface of the mould. **c**, **d** The composite material (NdFeB+Ecoflex) is filled and remains in the high-aspect-ratio hole structures. Pure Ecoflex is poured on top as the flexible substrate. The mould is placed in a hot oven and Ecoflex is fully cured. **e**–**g** After it is cooled down, the carpet is peeled from the mould structure. The soft cilia carpet is wrapped around a template and magnetized in the magnetizer ($B = 1.2$ T). **h** The magnetic cilia carpets with encoded magnetization information show metachronal waves in low magnetic fields ($B = 80$ mT).

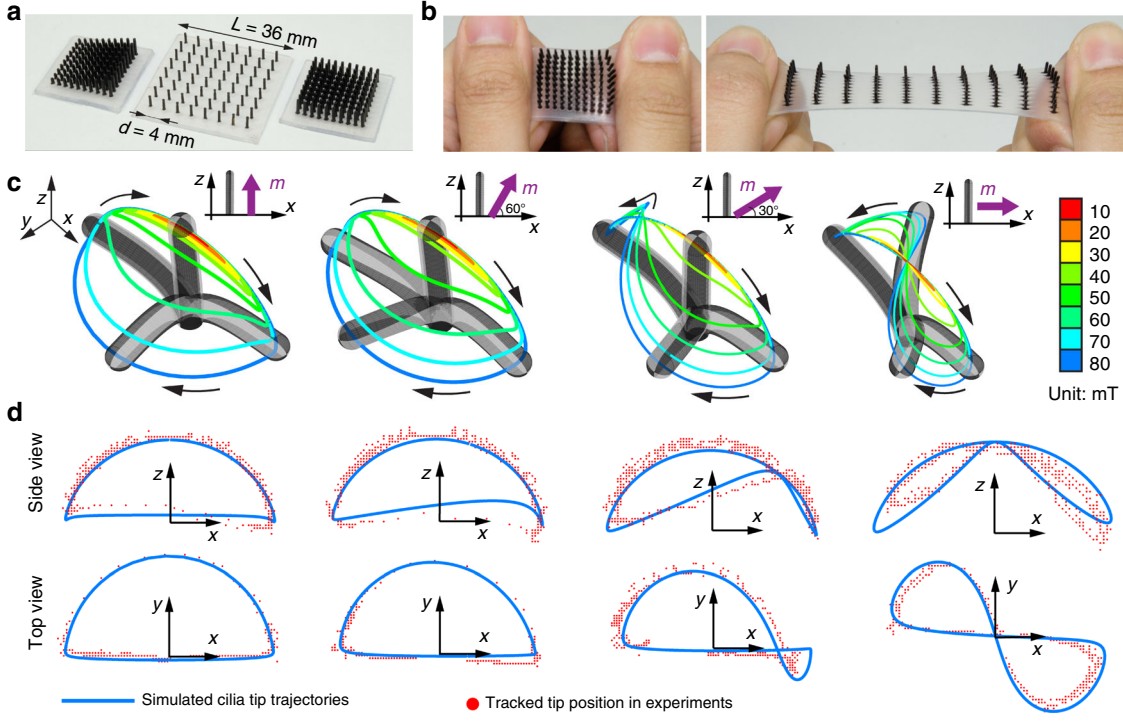

**Fig. 2 Stretchable cilia carpets and motion of a single cilium. a** Top views of different designs of magnetic cilia carpets. The cilia carpets are highly customizable with varying cilia lengths and positions. The left sample has an increasing hair length on one carpet. **b** Stretching the soft cilia carpets by hand. The soft substrate (Ecoflex 00-30) provides excellent stretchability to various curved surfaces. **c** Simulation results of a magnetic cilium under a rotating magnetic field in the *x*–*z* plane. The tip trajectories are shown with coloured lines under various magnetic field strengths. From left to right, four different magnetizations of identical cilia hair are shown. **d** Comparison of simulation and experiential tracking data of the artificial cilia with different magnetization with 80 mT rotating magnetic field in the *x*–*z* plane.

To explain magnetic cilia motion, we built a magneto-elastic model of cilia under a known magnetic field in quasi-static conditions. The model is based on Cosserat rod theory, which is suitable for simulating the elastic behaviour of slender beam-like soft magnetic structures[40]. The determination of the cilia's equilibrium set is performed using a numerical continuation framework[41]. The magnetic cilia are driven by a rotating magnetic field produced by a system composed of eight current-controlled electromagnets that are calibrated before the experiments[42]. Our simulation results match the experimental results as shown in Fig. 2d and Supplementary Fig. 1.

We find non-reciprocal cilia trajectories with increasing magnetic field strengths. For small magnetic field strengths (10–40 mT), the magnetic cilia exhibit reciprocal motion with a certain periodicity, as shown in Fig. 2c. From 50 to 80 mT, the magnetic cilia show non-reciprocal out-of-plane biomimetic trajectories in 3D, although the actuation field is in the x–z plane. In low Reynolds numbers ($Re \ll 1$), this non-reciprocal motion is crucial for natural cilia to pump liquid in their vicinity[2,11], according to the Purcell's "Scallop Theorem"[43].

Another finding is cilia with different magnetization directions have different trajectories. Here we study artificial cilia with four selected magnetization directions (0°, 30°, 60°, and 90° with respect to the z-axis in the x–z plane). The cilium with the magnetization along the z-axis exhibits biomimetic "D"-shaped trajectories, showing strong similarities with natural cilia and the most efficient cilium[44]. For a cilium with magnetization along its short axis, the tip trajectory is "8"-shaped. This is due to the elastic rotation of the hair (Supplementary Movie 2). The tracking results match our model as in Fig. 2d with both side and top views. Our model and experimental results provide a library of cilia hair trajectories enabling us to program subtle differences between neighbouring cilia in a large cilia carpet (Supplementary Fig. 2).

**Programmable metachronal waves.** Depending on the wave propagating direction and cilia beating direction, there are four basic metachronal waves (symplectic, antiplectic, dexioplectic, and laeoplectic)[45], defined by E. W. Knight-Jones in 1959. In symplectic and antiplectic metachronal waves, the wave directions are the same or opposite to the cilia power stroke direction, respectively. Dexioplectic and laeoplectic, also known as diaplectic metachronal waves, have wave directions perpendicular to the power stroke. In our magnetic cilia system, the power stroke is determined by the rotating magnetic field direction (in the x–z plane as in Fig. 2c). By using the magnetization templates with different curvature, we can encode different metachronal wavelengths across the cilia carpets for both symplectic and antiplectic waves (Fig. 3, Supplementary Fig. 6, and Supplementary Movies 1 and 7). By applying a rotating magnetic field in different planes, standing diaplectic waves are achieved on the same cilia carpet. As in Supplementary Movie 3, combinations of antiplectic and diaplectic waves are possible by simply changing the rotating magnetic field directions.

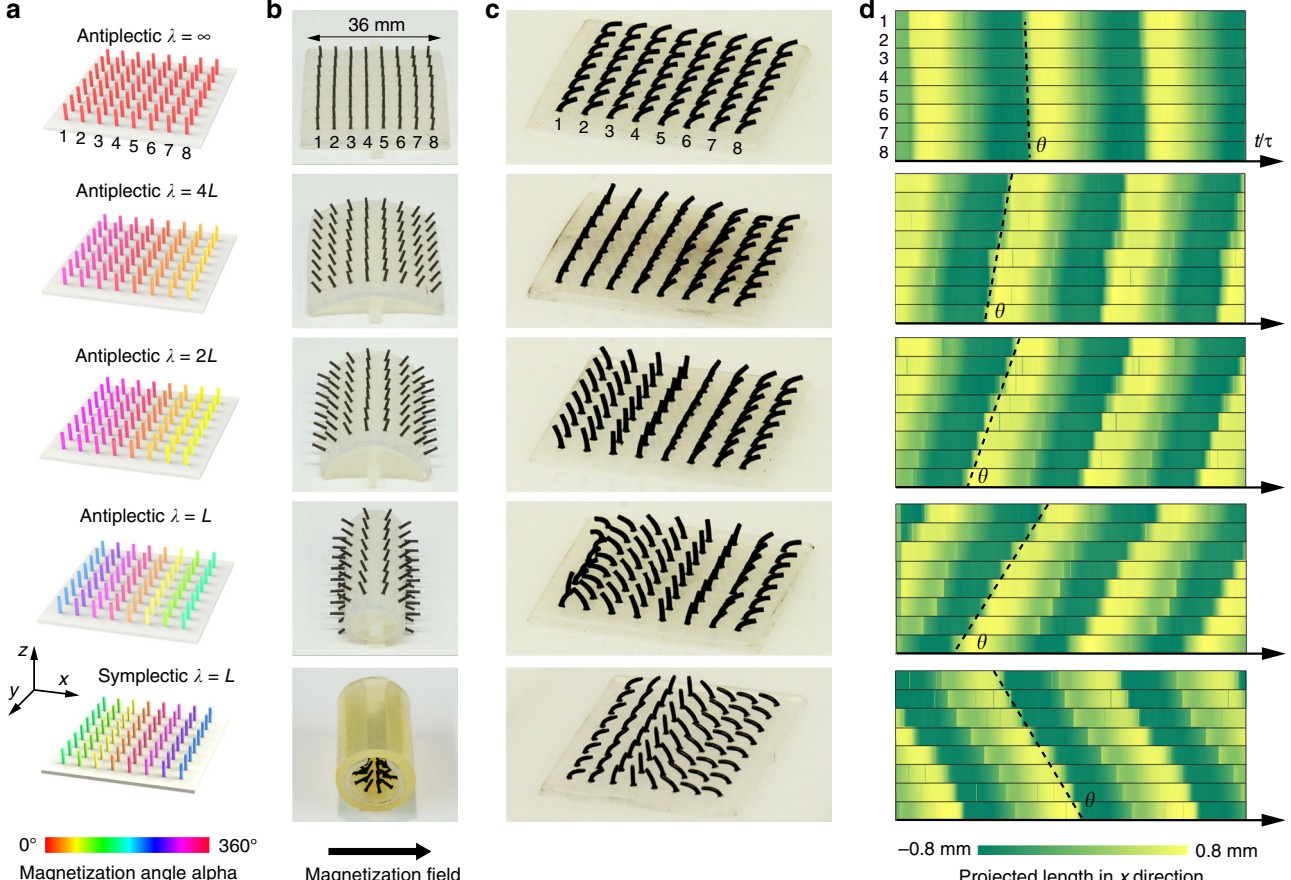

**Fig. 3 Programmable metachronal waves on magnetic cilia carpets. a** Targeted magnetization patterns of cilia carpets with different metachronal wavelengths. **b** 3D printed templates with different curvatures to encode the targeted magnetization patterns. The cilia carpets are wrapped on the surface of the templates, and a large magnetic field pulse (1.2 T) is applied in the magnetizer. **c** Top view of the 8 × 8 cilia carpets under dynamic magnetic fields. **d** The phase fields of metachronal waves propagate over time. The cilia hair motions are tracked based on their projected length in the x-direction. Lines of cilia with different magnetizations are numbered. The slope angle θ represents the number of metachronal wavelengths across the carpets.

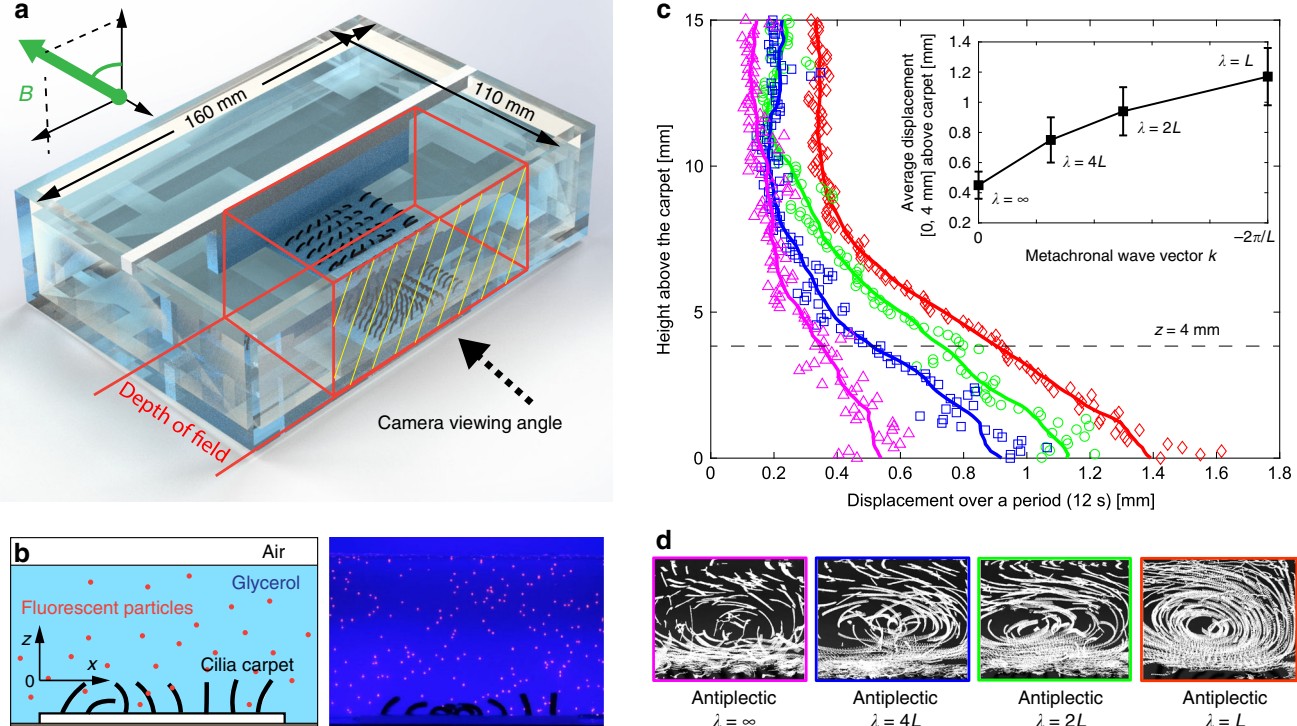

**Fig. 4 Metachronal waves promote liquid transport on cilia carpet. a** Experimental set-up of the fluidic transport on cilia carpets. The workspace is illuminated by blue light LEDs and the fluorescent particles (200 μm) on top of the cilia are in focus and their positions tracked. **b** Schematic view and experimental image. **c** The average displacement of the fluorescent particles on cilia carpets with different metachronal wavelengths. The coloured lines are smoothed results of the nearest 21 points. The average displacement over a period is calculated for particles between 0 and 4 mm above the cilia carpet, as shown in the inset figure. Error bars represent the s.d. of the measurements. **d** The actual trajectories of the particles and the flow pattern of different wavelengths cilia carpets. The plot area is 36 mm large and covers all the cilia carpet and 27 mm high between the tip of the cilia and the liquid surface.

**Fluid transport properties of cilia carpets**. The capability of promoting fluid transport using metachronal waves is one of the main functions of cilia carpets in tissues, from the respiratory tract to the oviduct, and an artificial cilia carpet should be able to mimic this important property. In the literature, metachronal waves are known for enhancing fluid transport on cilia carpets compared to identically beating cilia (Supplementary Table 1). Detailed discussions about the mechanism can be found in these studies[5,17]. Here we demonstrate the capability to study cilia–fluid interactions experimentally using our soft robotic system. Our aim is to experimentally test two important numerical results recently obtained, i.e. the threefold increase in propulsion velocity due to the formation of a metachronal wave[5] and a density dependency in the propulsion velocity[17].

In the experiments, the non-reciprocal motions of the magnetic artificial cilia (at 80 mT) promote liquid transport near the cilia carpet. The use of high viscosity liquid (99% glycerol) and low speed rotating magnetic field (30 degrees/s) ensure the low Reynolds number environment during the experiments, mimicking the hydrodynamic environment of natural cilia at the micrometre scale. We tracked the trajectories of the fluorescent particles and quantified the displacements over a period (12 s). The tracking region is a 40 mm × 30 mm rectangle from the tip of the cilia to the liquid surface (Fig. 4b; Supplementary Figs. 4, 8, and 9; and Supplementary Movie 4). We plot the average displacement over height, as shown in Figs. 4c and 5d. We also compared different measures of the transport of the tracer particles in Supplementary Fig. 5. In order to ease the comparison between different geometries, we also calculated the average displacement in the 4 mm of fluid above the carpet (Fig. 4c, inset and Fig. 5d, inset).

Figure 4 shows that the $\lambda = L$ sample has three times the average displacement within a period compared to the infinite-wavelength carpet. The presence of a metachronal wave promotes the formation of the coherent flow structure (Fig. 4d), showing more efficient liquid transport. Our experimental results strongly support numerical results with similar configurations[5]. In addition to wavelength, another parameter influencing pumping efficiency is the cilia density. Owing to the flexibility of our system, we fabricate cilia carpets with different distances between neighbouring cilia. We observe the average displacements follow the trend $v \sim d^{-a}$ with the exponent $a = 1.5$, when the distance $d$ is varied between 4 and 10 mm. Remarkably, the same trend is found in both numerical studies[5,17] ($a = 1.4$ and $a = 1.6$, respectively). However, in our experimental systems, the displacement of the tracking particles increases until the distance between neighbouring cilia is so small that the cilia cannot perform full trajectories, due to the strong magnetic and mechanical interactions between neighbouring cilia (Fig. 5b). This crowding problem has been identified as a limiting factor in the previous studies[5].

**Soft robotic locomotion using metachronal waves**. Magnetic soft robots exhibit great potential for biomedical applications inside the human body[46,47], especially inside the gastrointestinal tract[48–51]. Here we use magnetic cilia carpets (4 mm length, 0.8 mm diameter, 4 mm periodicity in the array) and demonstrate their potential as mobile soft robots, which use travelling metachronal waves to generate the locomotion gait. The soft cilia carpet is inverted and placed on a rigid plastic surface in a magnetic actuation system. The soft substrate of all cilia carpets is composed of 1-mm thickness pure Ecoflex 00-30. Unless

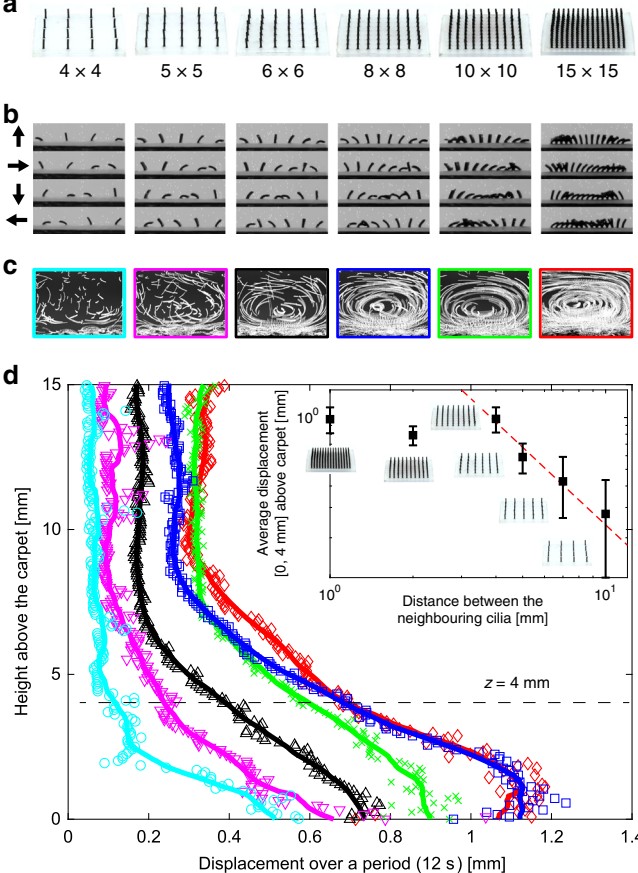

**Fig. 5 Pumping performance with cilia densities. a** Images of soft cilia samples with different cilia densities. The distance between the neighbouring cilia is 10 mm (4 × 4 carpet), 7 mm (5 × 5 carpet), 5 mm (6 × 6 carpet), 4 mm (8 × 8 carpet), 2 mm (10 × 10 carpet), and 1 mm (15 × 15 carpet), respectively. **b** Snapshots of cilia carpets with different densities under a magnetic field (80 mT) with various directions during the pumping experiments. The black arrows on the left side indicate the magnetic field directions within a cycle. Metachronal waves can be achieved on all samples with different cilia densities. **c** The trajectories of the tracking particles on top of the cilia. The plot area is 36 mm large and covers all the cilia carpet and 27 mm high between the tip of the cilia and the liquid surface. **d** The average displacement of tracking particles over a period. The coloured lines are smoothed results of the nearest 21 points. Error bars represent s.d. of the measurements.

specified, a uniform magnetic field is rotating clockwise at the speed of 30 degrees/s.

We found two modes of locomotion of the cilia soft robot, namely crawling and rolling. The mode of locomotion depends on the strength of the magnetic field. In the crawling mode, the soft robot locomotes thanks to the travelling metachronal waves of the cilia structures. With a magnetic field >60 mT, the magnetic torque is strong enough to curl up the carpet and allow the soft robot to roll forward (Fig. 6, Supplementary Fig. 7, Supplementary Movie 5). The transition can be explained by the balance among elasticity, gravity, and magnetic torque on the soft substrate of the cilia carpet. The rolling motion is significantly faster than the crawling motion for an 8 × 8 cilia carpet.

Inspired by the giant African millipede, we made a long cilia carpet with 20 × 5 cilia and encoded antiplectic waves with a wavelength $\lambda = 6d$ (Fig. 7c). Similar to the giant African millipede, the legs of the millipede-inspired soft robot are organized in metachronal waves and move forward (Fig. 7d, Supplementary Movie 6). We also studied the influence of the metachronal wave vectors on the walking speed of millipede-inspired soft robots. By folding the soft robot into different magnetizations templates (Fig. 8a), we encode antiplectic and symplectic waves with various wavelengths into the soft robots. We also test a soft robot of identically beating cilia without any metachronal wave. As in Fig. 8b and Supplementary Movie 8, soft robots with antiplectic waves exhibit much higher locomotion speed than the ones with symplectic waves. We think this difference can be explained by the opposite curvatures of the carpet substrate during the recovery stroke. For soft robots with symplectic waves, the substrate dents during the recovery stroke. This prevents the magnetic cilia to swing back by introducing additional friction with the surface. On the contrary, antiplectic soft robot bugles during the recovery stroke and allows the cilia to move forward efficiently, as in Fig. 8c.

## Discussion

We present a soft robotic system of magnetically actuated artificial cilia carpets. Large arrays of artificial cilia can be easily fabricated by curing magnetic composite material in the 3D printed moulds. Both symplectic and antiplectic metachronal waves with various wavelengths can be encoded into the artificial cilia carpets using curved templates in the magnetizer. In this work, the dimensions of the moulds are determined by the resolution of the 3D printer, which sets limitations for the cilia size and the distance between neighbouring cilia. Moulding magnetic artificial cilia at micrometre scale has been realized using SU-8 patterned by photolithography[52]. Increasing NdFeB particle concentration in the composite material will increase the viscosity of the mixture, making it more difficult to process. Magnetic actuation system (CardioMag) is bulky and slow in dynamic responses. The fluid pumping experiments support previously published numerical results, showing the platform's ability to study complex cilia–fluid interactions at the micrometre scale. We hope this highly customizable robotic platform could assist in studying the collective behaviours of natural cilia and cilia-based active matter systems.

## Methods

**Fabrication of soft cilia carpets.** The polymer moulds are printed using Vero-Clear Material (Stratasys Inc.). After printing, the moulds are cleaned using a water jet to remove all supporting materials and washed with soap several times. Then the mould is cured with ultraviolet in the FormCure (Formlabs Inc.) at 80 °C for 8 h. Later, the moulds are cleaned with isopropyl alcohol and dried using an air gun. A thin layer of resin (XTC-3D, Smooth-On Inc.) is brushed on surfaces of the mould and cured in an oven at 65 °C for 1 h.

In the first moulding step, the NdFeB microparticles (MQP-S-11-9, Magnequench) and silicon rubber (Ecoflex 00-30, Smooth-On Inc.) are mixed in a 1:1 weight ratio. The mixture is placed in the vacuum chamber for 5 min and pushed into the mould to fill the cilia volume. The excess composite is removed. In the second moulding step, the pure Ecoflex (Ecoflex 00-30, Smooth-On Inc.) is poured in the mould to fill the carpet volume. The filled moulds are placed in an oven at 65 °C for 8 h. After cooling, the magnetic cilia carpet is peeled off the mould by hand. Owing to the soft nature of Ecoflex, it is easy to de-mould without damage to the cilia carpet. An impulse magnetizer (IM-10, ASC Scientific) with maximum 1.2 T is used to magnetize all samples.

**Simulations of the magnetic cilia.** The cilium is assumed to be subjected to gravity and magnetic forces and torques induced by externally generated magnetic fields in quasi-static conditions. In addition, internal stresses are assumed to be due only to elastic deformation. Magnetization is assumed to be permanent and homogeneous throughout the cilia body. The material of the cilia is assumed to be linearly elastic and isotropic.

The cilium is modelled as a flexible magnetic rod with a constant cross-section, which state along its length $L$ is parametrized by the arc length $s$. At the station $s$, the state of the rod is fully described by its position $\mathbf{p}(s)$, orientation quaternion $\mathbf{Q}$ $(s)$, internal force $\mathbf{f}(s)$, and internal torque $\boldsymbol{\tau}(s)$. The quaternion is a $4 \times 1$ vector written as $\mathbf{Q} = [q_r \ \mathbf{q}]^T$ (dependency $s$ is not reminded in the rest of the section). The corresponding rotation matrix is denoted $R(\mathbf{Q})$.

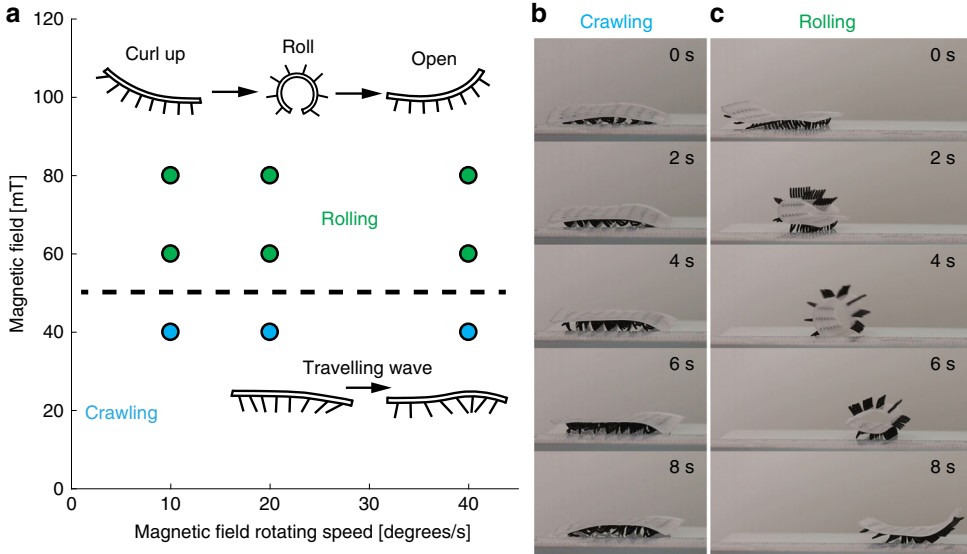

**Fig. 6 Multi-modal robotic locomotion. a** Based on the operating magnetic field strength, we find two distinct modes of locomotion, rolling and crawling. **b** With a low magnetic field (40 mT), the travelling metachronal wave allows the soft robot to crawl on a hard surface. **c** For magnetic field strengths >60 mT, magnetic torque overcomes gravity and the elastic energy in the soft carpet. The soft robot can curl up and roll to move forward (Supplementary Movie 5).

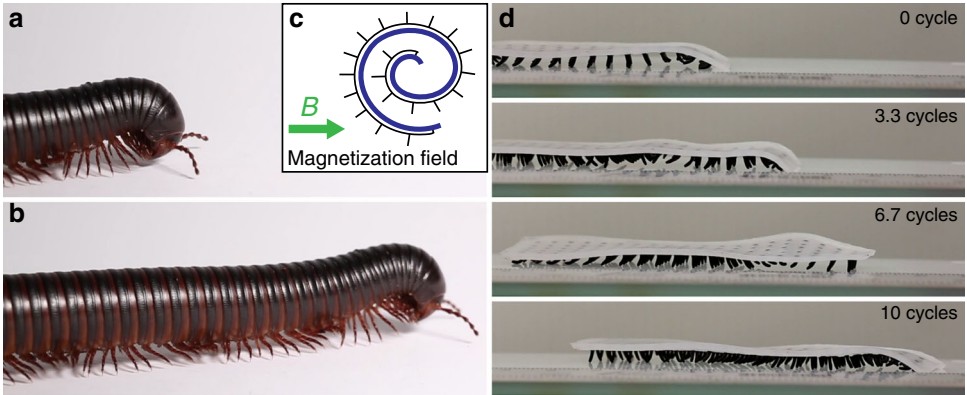

**Fig. 7 Millipede-inspired soft robot crawl using metachronal waves. a**, **b** A crawling giant African millipede. The legs are organized as travelling metachronal waves. **c** The magnetization template to magnetize the soft cilia carpet (20 × 5 cilia array) in the magnetizer. Two wavelengths are encoded. **d** Experimental results of the crawling soft robot inspired by the giant African millipede (Supplementary Movie 6).

The evolution of the state of the cilium along $s$ is described by Cosserat rod theory as previously introduced for soft magnetic rods[40]. Cosserat rod equations describe the kinematics of the cilia along the arc length as follows

$$\frac{d\mathbf{p}}{ds} = R(\mathbf{Q})(\mathbf{K}_{se}R(\mathbf{Q})^{T}\mathbf{f} + \mathbf{v}^{*})$$

$$\frac{d\mathbf{Q}}{ds} = \frac{1}{2}\begin{bmatrix} -\mathbf{q}^{T} \\ q_{r}\mathbf{I} - [\mathbf{q}]_{\times} \end{bmatrix} R(\mathbf{Q})(\mathbf{K}_{bt}^{-1}R(\mathbf{Q})^{T}\boldsymbol{\tau} + \mathbf{u}^{*})$$

$$\frac{d\mathbf{f}}{ds} = \frac{-(\nabla(R(\mathbf{Q})\mathbf{m} \cdot \mathbf{B}(\mathbf{p})) - \rho V \mathbf{g})}{L}$$

$$\frac{d\boldsymbol{\tau}}{ds} = [\mathbf{f}]_{\times}\frac{d\mathbf{p}}{ds} - \frac{[R(\mathbf{Q})\mathbf{m}]_{\times}\mathbf{B}(\mathbf{p})}{L}$$

where $[\cdot]_{\times}$ denotes the skew-symmetric matrix form of the vector cross product, $\mathbf{I}$ is the identity matrix, $\mathbf{v}^{*}$ is the direction of the centreline tangent of the rod (in our case, we choose $\mathbf{v}^{*} = [0\ 0\ 1]^{T}$), $\mathbf{u}^{*}$ is its intrinsic curvature, $\rho$ is the volumetric mass density of the cilium, $V$ is its volume, $\mathbf{m}$ is the magnetic moment of the cilium in its straight configuration, $\mathbf{B}(\mathbf{p})$ is the magnetic field acting at position $\mathbf{p}$, and $\mathbf{g}$ is the gravitational constant.

$\mathbf{K}_{se} = \text{diag}(GA, GA, GA)$ and $\mathbf{K}_{bt} = \text{diag}(EI_{x}, EI_{y}, EI_{z})$ are the 3 × 3 shear/extension and bending/torsion stiffness matrices, respectively, with Young's modulus $E$ of the cilium material, $G$ its shear modulus, $A$ the cross-section area, and $I_{x}$, $I_{y}$, $I_{z}$ the principle moments of area.

The magnetic field $\mathbf{B}(\mathbf{p})$ is externally generated by an electromagnetic navigation system (eMNS) composed of eight current-controlled electromagnets. The eMNS is modelled as a linear multi-source dipole so that the field $\mathbf{B}(\mathbf{p})$ can be estimated as $\mathbf{A}(\mathbf{p})\mathbf{i}$, where $\mathbf{i}$ is an 8 × 1 vector of the electromagnets currents and $\mathbf{A}(\mathbf{p})$ is a 3 × 8 actuation matrix that has been calibrated[42].

The simulations are performed assuming that the reference magnetic field $\mathbf{B}^{*} = \mathbf{B}(\mathbf{p}_{0})$ is generated at the centre $\mathbf{p}_{0}$ of the eMNS with the current vector $\mathbf{i}^{*}$ to produce a rotating magnetic field of the form $\mathbf{B}^{*} = [B\sin(\theta)\ 0\ B\cos(\theta)]^{T}$ with $\theta \in [0, 2\pi]$ in the cilium reference frame and for a magnitude $B$ varying between 10 and 80 mT. The current vector $\mathbf{i}^{*}$ is computed as $\mathbf{A}^{+}(\mathbf{p}_{0})\mathbf{B}^{*}$ with $\mathbf{A}^{+}$ the Moore-Penrose pseudoinverse of $\mathbf{A}$, and the field at $\mathbf{p}$ as $\mathbf{B}(\mathbf{p}) = \mathbf{A}(\mathbf{p})\mathbf{i}^{*}$. The cilium magnetization is chosen as $\mathbf{m} = m[\sin(a)\ 0\ \cos(a)]^{T}$ with $a \in \{0, 30°, 60°, 90°\}$. The parameters chosen for the simulations are summed up in Table 1.

The magnetization magnitude is estimated considering that the NdFeB particles are mixed with Ecoflex 00-30 with weight ratio 1:1 and have a residual flux density of $7.40 \times 10^{-1}$ T. The Young's modulus is identified with a standard tensile test, and the shear modulus is estimated considering incompressibility property of the material (Poisson ratio of 0.5). The rest of the parameters are found in the material supplier datasheet.

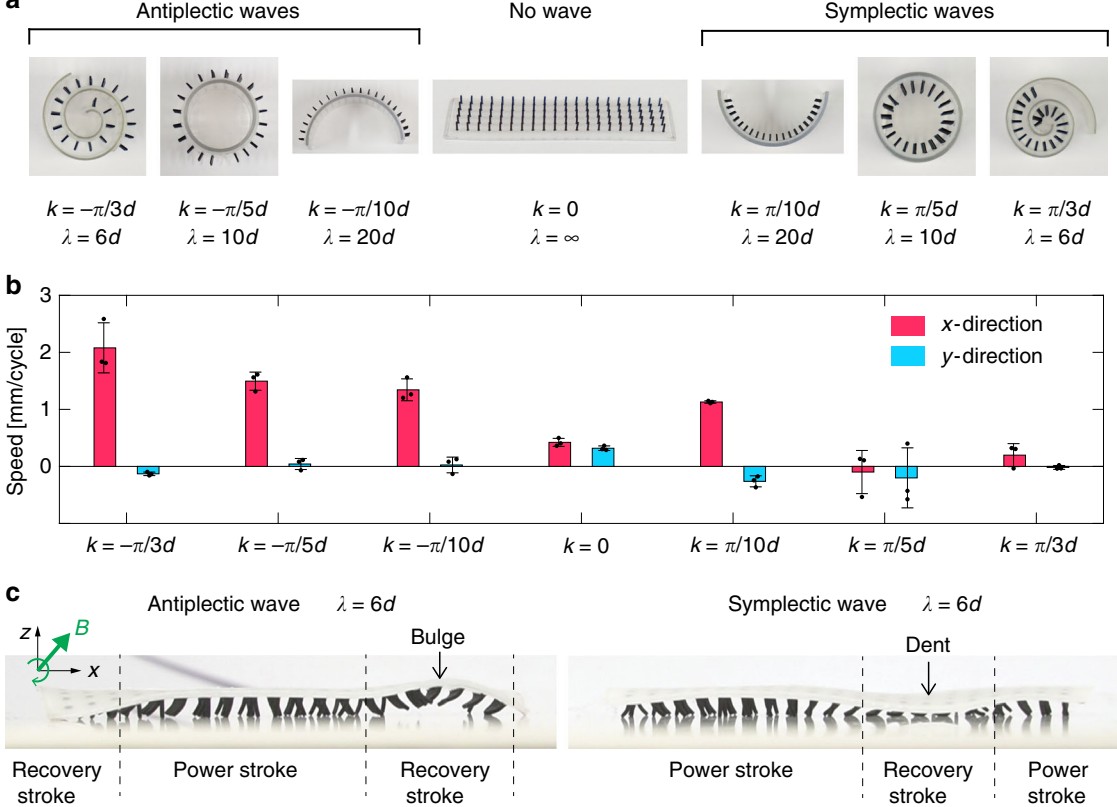

**Fig. 8 Influence of metachronal wave vectors on the crawling speed of soft robots. a** The magnetization templates that encode antiplectic and symplectic waves on the cilia soft robots. The wave vector and wavelengths are denoted $k$ and $\lambda$, while $d$ is equal to 4 mm, which represents the distance between the neighbouring cilia. **b** Walking speeds of soft robots with different metachronal waves. The locomotion speeds are measured based on the geometric centre of the soft robots. The error bars represent the s.d. of three measurements. **c** Curved substrate on antiplectic and symplectic wave soft robots. The substrate of antiplectic soft robot bulges in the recovery stroke, helping the legs to move forward. On the contrary, the substrate of the symplectic soft robot dents and obstructs the recovery stroke, which slows down the robotic locomotion.

**Table 1 Summary of the parameters used for simulations of single magnetic cilium.**

| Parameter | Notation | Value | Unit |
|---|---|---|---|
| Magnetization magnitude | $m$ | $3.82 \times 10^{-5}$ | A·m$^2$ |
| Young's modulus | $E$ | $1.85 \times 10^{5}$ | Pa |
| Shear modulus | $G$ | $6.16 \times 10^{4}$ | Pa |
| Volumetric mass density | $\rho$ | $2.39 \times 10^{3}$ | kg/m$^3$ |
| Length | $L$ | $4.00 \times 10^{-3}$ | m |
| Diameter | $d$ | $8.00 \times 10^{-4}$ | m |

The system of equations is further discretized in 20 sections using finite differences and solved using MATCONT[53], which is a numerical continuation framework that can be used to follow the mechanical equilibrium branches of flexible magnetic rods, as previously proposed in ref. [41].

**Fluid transport experiments**. The fluidic transport experiments are performed in an 8-coil electromagnetic system (CardioMag) with a calibrated cube workspace (20 cm × 20 cm × 20 cm) in the centre[40]. A uniform magnetic field of 80 mT is applied in the workspace with a rotating speed of 30 degrees/s. The cilia carpet is placed in a 160 mm large, 110 mm wide, and 45 mm deep acrylic box. The observation window is the 160 mm large side of the box. In the middle width of the box (55 mm from the observation window), we place a 90 mm large and 45 mm high white polyoxymethylene (POM) board. The square cilia carpet (36 mm × 36 mm) is placed between the observation window and the POM board. In all, 99% glycerol (ABCR Inc., viscosity: 1.15 Pa·s) is mixed with small amounts of red fluorescent particles (Copheric Inc., diameter: 200 μm), and the box is filled up to

30 mm. In this way, only the liquid flow on the top of the cilia carpet is captured by the digital camera (X-T20, Fijifilm Inc.) at the same height. The video is then processed using a customized MatLab script to track the trajectories of the fluorescent particles.

**Particle trajectory analysis**. We tracked the trajectories of the fluorescent particles and quantified their displacements over a period (12 s). We constructed maps of displacements by assigning the displacement over a period to the $x$ and $z$ coordinates averaged over the same period (Figs. 4d and 5c) and vertical profiles of displacements by averaging the maps along the $x$-direction over the entire length of the carpet (36 mm) (Figs. 4c and 5d). To ease the comparison between different geometries, we calculated the average displacement in the 4 mm of fluid above the carpet (Fig. 4d, inset and Fig. 5d, inset). This average particle displacement within a period is used to evaluate the pumping performance of the cilia carpet. The depth of field of our optical system covers the entire fluidic set-up (Fig. 4a). Consequently, all the particles moving on top of the cilia carpet are on focus and displacements maps are averaged over the width of the carpet.

**Soft robotic locomotion**. The cilia-inspired soft robots are placed and actuated in the same eight-coil electromagnetic system. A POM board is placed and fixed at the centre of the workspace. The surface of the POM board is smooth and dry. We use a thin layer of baby powder on the soft robots to prevent unwanted adhesion. Then the cilia carpet is placed on top of the POM board and a clockwise rotating magnetic field is applied in the $x$–$z$ plane. For the millipede soft robot, a uniform magnetic field with 30 degrees/s is applied to drive the travelling metachronal wave locomotion. The motions of the cilia-inspired soft robot are recorded using digital cameras (Fujifilm X-T20, Sony Alpha 6400) from top and side views.

## Data availability
The authors declare that data supporting the findings of this study are available within the paper and its Supplementary Information files.

## Code availability

All the relevant code used to generate the results in this paper and Supplementary Information is available upon request.

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

## Acknowledgements

This work is financially supported by ETH grant (1916-1), SNSF project funding (185039), and ERC grant SOMBOT (743024). E.S. and G.S. acknowledge support from SNSF PRIMA grant (179834). We thank Jung-Chew Tse for assistance in 3D printing, Christoph Chautems for his assistance using the CardioMag, Penfei Liu for introduction to the magnetizer, and Stuart Hamilton for the source of the giant African millipede walking video.

## Author contributions

H.G. conceived the project and managed the research. H.G. and H.C. fabricated the cilia carpets. H.G., H.C., and Q.B. performed the experiments of magnetic cilia actuations. E.S. and G.S. performed the particle tracking velocimetry. Q.B. and Q.P. performed the cilia hair simulations. C.D.M., S.G., T.-Y.H., S.P., A.H., and D.A. participated in discussions. H.G., Q.B., and E.S. wrote the manuscript with contributions from all authors. B.J.N. supervised this project.

## Competing interests

The authors declare no competing interests.
