## [Peer Review File · Nature Communications]

Reviewers' comments:

Reviewer #1 (Remarks to the Author):

This manuscript develops a method to fabricate programmable magnetic cilia carpet with a large number of cilia hair structures. Complex magnetization pattern can be encoded in the cilia arrays, facilitating more intensive investigation on motile cilia. Using this method, the authors succeed to develop a pumping device and cilia soft robot, which exhibit great potential for cilia-based active matter systems and robotic applications. Therefore, this work is recommended for publication in Nature Communications if the following questions can be well addressed.

1. As the method provides a customized fabrication platform for programmable cilia, more fundamental and detailed studies are recommended to investigate the influence of cilia aspect ratio, elastic modulus and magnetization direction on their bending angle under a specific magnetic field.
2. In line 125, the authors said "Our simulation results match the experimental results as shown in Figure 2d and supplementary figures". But I cannot find the supplementary figures.
3. The authors should discuss the reasons why increasing the wavelength of metachronal waves can increase the pumping performance.
4. What is the influence of cilia length on the pumping performance? For cilia carpets with an increasing length (The left sample in Fig. 2a) and identical length (The right sample in Fig. 2a), which one has better pumping performance?
5. What is the influence of cilia wavelength on the locomotion speed for the millipede-inspired soft robot?
6. The manuscript is lack of a conclusion part.

Reviewer #2 (Remarks to the Author):

The paper describes magnetically actuated cilia-like post arrays that exhibit metachronal motion. Whereas similar polymeric posts with embedded magnetic particles have been previously demonstrated, the novelty of the work in differential magnetization of the posts leading to spatially varying magnetic force under a uniform magnetic field. This in turn generate metachronal motion in the post array. The paper demonstrates the applications of such arrays capable of metachronal motion to fluid pumping and robotic locomotion.

1. There is limited information about the properties of the magnetic posts. I was only able to find that they are 4 mm in length in SI. Information about post dimensions and other relevant properties need to be in the main text. Magnetic particle size needs to be also reported.
2. Comparing to previously published magnetic cilia, the authors claim that the main advantage of their approach is the simplicity of the fabrication method. While this could be arguably true, for practical applications much more important criteria are dimensions of cilia, actuation amplitude, and frequency. These metrics should be included and discussed to position this work with respect to previous publications.
3. Dimensions of the setup for fluid transport experiment are missing. The results are confusing. Previous works showed that velocity above ciliary arrays changes linearly with height. Here, however, velocity decreases to a constant value some distance about cilia (Fig. 4c and Fig. 5d). What is the reason for this constant velocity and why it depends on the wavelength? How was it ensured that the fluorescent particle displacement is not affect by the initial transient and the flow reached steady circulation? It does seem that the experiments were not performed sufficiently accurate.
4. The velocity of fluid seems to be rather slow. In Fig. 4c and Fig. 5d, it will be more useful to present the velocity data normalized by the length of posts and oscillation period. Furthermore, the results need to be compared to previously published experiments on pumping using artificial cilia.

5. In biological systems, metachronal waves often associated with hydrodynamic coupling between individual cilia. Are there any effects of the fluid on the motion of the magnetic cilia and the wave propagation? How pumping velocity depends on the beating rate? The change of the pumping rate with the frequency need to be explored.
6. The authors claim that metachronal waves are beneficial for fluid pumping. However, there is no discussion of any physical mechanism enhancing fluid pumping to support this. Experiments are also not convincing due to the problems pointed in 3.
7. The locomotion is interesting but very briefly investigated. The traveling speed of the structure and its relations to the wavelength and speed, array density need to be reported.
8. The limitations of the proposed method need to be discussed. It will be especially useful to discuss how these cilia can be miniaturized.

Reviewer #3 (Remarks to the Author):

The manuscript reports a detailed experimental study of the dynamics and transport properties of magnetically actuated, soft, artificial cilia carpets. A new method is presented to fabricate soft, magnetic, cylindrical filaments of sub-millimeter diameter and a few millimeter length, anchored on a deformable membrane substrate. Programmable magnetization patterns can be encoded into these artificial cilia carpets, which then display metachronal wave-like beat patterns in dynamic magnetic fields. The experiments are accompanied by numerical simulations based on elasticity theory. Both transport in a fluid environment and the locomotion (of an "inverted" carpet) on a solid surface have been investigated. The system provides a highly customizable experimental platform. It has been used here to test previous numerical predictions for the efficiency of metachronal transport and the dependence of transport velocity on cilia density. The latter dependence has been difficult to study experimentally so far, due to the difficulty in fabricating samples with various cilia densities.

I think that the current study presents a very significant step forward, both in the fabrication and control of soft robots, and the experimental study of the physics of metachronal dynamics and transport.

Therefore, I support the publication of the manuscript in Nat. Comm., after the authors have considered the following questions and comments:

- (1) What is the length of the cilia?
 - (2) There is one big difference between metachronal waves in artificial and biological systems, which is that these waves are actuated in the former, but self-organized in the latter system. This has apparently (and not unexpectedly) little effect on the transport properties. Nevertheless, it would be good to point out this difference clearly.
 - (3) Why does the longer power-stroke phase make the metachronal waves antiplectic?
 - (4) I find the notation zero wavelength, quarter wavelength, etc. in Fig. 4 confusing. I guess this means infinite wavelength, wavelength of 1/4 of systems size, etc. Please clarify.
 - (5) The section of "soft robotic locomotion" is very short and difficult to understand.
- What is the potential for biomimetic applications inside the HUMAN body?

The references give no indications of such approximations. Millimeter-size cilia seem to be quite large for such applications.

-- There is a rolling mechanism indicated at the top of Fig. 6a, which consists of a periodic curling and opening-up time sequence. What is the characteristic frequency? How does it depend on the magnetic field parameters? Why does the robot not simply stay permanently in the rolled-up state?

-- The authors state that "... locomotion ... does not solely rely on the individual legs, but instead on the wave pattern." This is surprising and interesting, since the waveform (symplectic, antiplectic, ...) makes little difference for fluid transport of cilia carpets. Do soft robots move in opposite directions for symplectic and antiplectic waves?

Reviewer #1 (Remarks to the Author):

This manuscript develops a method to fabricate programmable magnetic cilia carpet with a large number of cilia hair structures. Complex magnetization pattern can be encoded in the cilia arrays, facilitating more intensive investigation on motile cilia. Using this method, the authors succeed to develop a pumping device and cilia soft robot, which exhibit great potential for cilia-based active matter systems and robotic applications. Therefore, this work is recommended for publication in Nature Communications if the following questions can be well addressed.

We thank the reviewer for the positive evaluations of our work. We address the reviewer's comments point by point as follows.

1. As the method provides a customized fabrication platform for programmable cilia, more fundamental and detailed studies are recommended to investigate the influence of cilia aspect ratio, elastic modulus and magnetization direction on their bending angle under a specific magnetic field.

We appreciate the suggestion for detailed studies of the bending motions of magnetic artificial cilia. Indeed, scanning key parameters of the magnetic artificial cilia, including aspect ratio, elastic modulus, magnetization direction, and the external magnetic fields, could provide a useful library for those who would like to develop a customized magnetic artificial cilia system. Here, considering this large parameter space (elastic modulus, aspect ratio, magnetization, external magnetic field strength), we use the Cosserat rod model as a guide to understand the behaviors of magnetic artificial cilia and discuss these key parameters separately as follows. Although some materials are mentioned in the original submission, we would like to review them for a complete explanation.

Among these key parameters of cilia, we found that the cilia trajectories are significantly influenced by their magnetization directions. In Figure R1, we performed simulations (based on the Cosserat rod model) of the trajectories of artificial cilia within one period with different magnetic field strengths.

Figure R1, (identical to Figure 2c in the manuscript) Simulated cilia's tip trajectories under different magnetic field strength and directions. The Magnetization directions are shown with purple arrow in the insets. The external magnetic field is rotating in the x-z plane.

Figure R2 depicts the comparison between the simulated and measured trajectories of the cilia's tip for both side and top views along one cycle of a rotating magnetic field of 80 mT. In Figure R3, the simulation and tracked tip positions are compared in the time domain. These results show that our implementation of the Cosserat rod model is very useful to understand

the motion of the magnetic artificial cilia. The simulations show excellent agreement with the experiment, and capture the essential difference in the behavior of the cilia depending on their magnetization direction. This already allows us to simulate the motion of magnetic artificial cilia without experimentally testing all possible combinations of design parameters.

Figure R2, (identical to Figure 2d in the manuscript) Comparison of simulation and experimental tracking data of the artificial cilia with different magnetization and for a 80 mT rotating magnetic field in the x-z plane.

Figure R3, Artificial cilia's tip trajectories of the simulation results (blue curve) and experiments (red dots) for different magnetization directions. All experiments and simulations are performed under a rotating magnetic field of 80 mT inside the x-z plane. The magnetization directions are shown with large purple arrows. Unmatched data, where the tracked tip positions (red dots) are far from simulations (blue curve), is due to the limited tracking algorithm.

So far, we have studied the magnetization direction and the magnetic field strength. In the following paragraphs, we would like to argue that changing the cilia aspect ratio, elastic modulus, and magnetization strength are equivalent to changing the magnitude of the magnetic field. Considering that changing some parameters could significantly modify the fabrication process, we only access these parameters based on the model, and we try to provide an intuitive illustration of how these parameters could influence the artificial cilia dynamics.

- **A theoretical analysis using the Cosserat Model**

The four governing equations in the Cosserat Model are (as in the method)

$$(1) \quad \frac{d\mathbf{p}}{ds} = R(\mathbf{Q})(\mathbf{K}_{se}R(\mathbf{Q})^T\mathbf{f} + \mathbf{v}^*)$$

$$(2) \quad \frac{d\mathbf{Q}}{ds} = \frac{1}{2} \begin{bmatrix} -\mathbf{q}^T \\ q_r\mathbf{I} - [\mathbf{q}]_{\times} \end{bmatrix} R(\mathbf{Q})(\mathbf{K}_{bt}^{-1}R(\mathbf{Q})^T\boldsymbol{\tau} + \mathbf{u}^*)$$

$$(3) \quad \frac{d\mathbf{f}}{ds} = \frac{-(\nabla(R(\mathbf{Q})\mathbf{m} \cdot \mathbf{B}(\mathbf{p})) - \rho V\mathbf{g})}{L}$$

$$(4) \quad \frac{d\boldsymbol{\tau}}{ds} = [\mathbf{f}]_{\times} \frac{d\mathbf{p}}{ds} - \frac{[R(\mathbf{Q})\mathbf{m}]_{\times}\mathbf{B}(\mathbf{p})}{L}$$

We represent the magnetization of the cilium \mathbf{m} , and the magnetic field \mathbf{B} as $\mathbf{m} = m_{mag}\mathbf{m}_0$ and $\mathbf{B} = B_{mag}\mathbf{B}_0$ to separate the magnitudes and the unit vectors. In equation (3) and (4), force \mathbf{f} and the torque $\boldsymbol{\tau}$ are calculated. One can easily factorize a scalar value $m_{mag}B_{mag}$ in the equations and consider it as a single parameter. This means changing the magnitude of the magnetic field is equivalent to changing the magnetization of the cilium, in the equations of motion.

Similarly, we can express the stiffness matrices \mathbf{K}_{bt} and \mathbf{K}_{se} in equations (1) and (2) based on the geometric parameters of the cylinder rod as $\mathbf{K}_{se} = \text{diag}(GA, GA, GA)$, and $\mathbf{K}_{bt} = \text{diag}(EI_x, EI_y, EI_z)$, where $A = \pi r^2$ is the area of the cross-section, $I_x = \frac{\pi}{4}r^4$, $I_y = \frac{\pi}{4}r^4$, $I_z = \frac{\pi}{2}r^4$ are the area momentum of inertia with r the radius of the cilium. If we assume that the material is incompressible, which is common for silicone elastomer, $G = \frac{1}{2} \frac{E}{(1+\nu)}$ where E and G are the Young's and the shear moduli respectively. This allows us to consider EI and GA as one parameter, where changing r^2 and E are equivalent regarding the bending of the artificial cilia. If the different system parameters (elastic modulus, aspect ratio, magnetization, magnitude of the magnetic field) combinations lead to identical four governing equations listed above, we can expect the cilia have the same behaviour.

- **Simulations**

We study the influence of the diameter and the elastic modulus on the motion of the magnetic cilium through our implemented Cosserat rod model. In the simulation, the Young's Modulus E is 185 MPa, the diameter of the cilium D is 0.4 mm, and its length

L is 4 mm. We applied a magnetic field of 80 mT in the positive Y direction, and the magnetization of the cilium is along the its long axis. The results are shown in Figure R4. We illustrate the fact that the distal bending angle of the cilium can be identical for different set of parameters, as depicted for the configurations marked with "A" to "C". In the case "A" for example, decreasing the diameter to D/2 is equivalent to decreasing the Young's modulus to E/4 from the nominal parameters.

Figure R4, Simulation results of cilia hair bending with different aspect ratios and different stiffness. In the simulations, we vary the diameter $D=0.8$ mm of the cylindrical hair and the Young's modulus $E=1.85 \times 10^5$ Pa.

We thus show that the implemented Cosserat rod model is useful to model the behavior of our magnetic artificial cilia. We also provide an intuitive explanation to navigate in the large

parameter space for customized cilia system. We hope our explanations are sufficient to address the comment from reviewer 1.

2. In line 125, the authors said "Our simulation results match the experimental results as shown in Figure 2d and supplementary figures". But I cannot find the supplementary figures.

We are sorry for this mistake, the matching results between simulations and experiments can be found in Figure R1-R3. We organized our supplementary figures and now you can find them in the supplementary material file.

3. The authors should discuss the reasons why increasing the wavelength of metachronal waves can increase the pumping performance.

This is a very good question. Metachronal waves' capability to promote pumping in cilia carpet is not a new discovery and it has been extensively reported in the literature from both theoretical [2]–[16] and experimental sides [13], [17]–[20]. Before answering the question, there are two things we would like to clarify.

First, we need to point out that we made a mistake on expression in Figure 3 as pointed out by reviewer 3. The metachronal wavelength of the cilia carpet is decreasing from infinite wavelength (identical beating cilia array) to wavelength $\lambda=8d=L$ (d is the distance between neighboring cilia). Consequently, our results show that decreasing metachronal wavelengths from infinity to carpet length(L) will increase the pumping speed.

Secondly, the decreasing wavelength will not always increase the pumping speed. In Figure 3 of a numerical study [13], the maximum pumping speed is measured at the metachronal wavelength $\lambda=4d=L/2$ ($k_x = \frac{\pi}{2d}, k_y = 0$), as in the figure R5. If the wavelength is smaller than $4d$, the pumping speed is decreasing with a decreasing metachronal wavelength.

[Redacted]

Figure R5, Previously published numerical studies of the relationship between the pumping speed of cilia carpet and metachronal wavevector k (reference Osterman PNAS). The black arrow represents the power-stroke direction, and the red arrow represents the wavevector. The pumping speed is plotted as a function of wavevectors represented with color in subfigure G (Red represents high pumping speed and purple represents low pumping speed).

In the literature, a few papers attempt to explain the physical mechanism of this phenomena. Here we summarize these explanations. In the paper from Osterman et. al. [13], authors explained that the cilia can stack densely and sweep back during the recovery stroke, which reduce the fluidic drag and backward flow. In [15], authors explained that the cilia arrays, which beat in metachronal waves, can exert their full force on the surrounding liquid compared with synchronized cilia arrays. In a numerical study [12], the authors proposed an explanation based on the vortex organization.

The mechanism is quite complicated and not fully understood yet. However, in this manuscript, we have added some comments and some references for the readers who are interested in this topic.

"In the literature, metachronal waves are known for promoting fluid transport on cilia carpets compared to identically beating cilia. Detailed discussions about the mechanism can be found in these studies. Here, we would like to demonstrate the capability to study cilia-fluid interactions experimentally using our soft robotic system."

4. What is the influence of cilia length on the pumping performance? For cilia carpets with an increasing length (The left sample in Fig. 2a) and identical length (The right sample in Fig. 2a), which one has better pumping performance?

We thank the reviewer for raising this very interesting question. Based on previously reported simulations and theoretical analysis, it is known that increasing the cilia length will increase the pumping speed, if other system parameters are kept the same. It can be understood as longer hair has a larger linear velocity at the tip with the same beating angular velocity. Based on this reasoning, we can conclude that longer hair will have higher pumping speed, so the left sample in Figure 2a will have a higher pumping speed than the right sample in Figure 2a.

In the literature, the cilia length is not often used as an independent parameter in terms of fluid pumping. Instead, the ratio between the cilia length and the distance between neighboring cilia (L/d) is often used. We follow this convention, already introduced in [13], [15], and vary the distance between neighboring cilia to study this parameter (L/d), as in Figure 5.

The purpose of the presented pumping experiments is to show that our enabling experimental system is able to benchmark experimentally for the first time previously published numerical results. For this reason, we think that using the parameter (L/d) will facilitate comparison with previous works and increase its impact on the community. More cilia design and arrangements can be fabricated and tested, thanks to the flexibility offered by the 3d printing technology. The formed is one of the main messages of the paper, which, for the sake of space, is not aimed at testing all the possible experimental parameters. We hope this work can inspire more detailed studies of cilia transport phenomena thanks to the design flexibility offered by our platform.

5. What is the influence of cilia wavelength on the locomotion speed for the millipede-inspired soft robot?

This is a very good question. We performed new experiments to investigate this topic using samples magnetized with different metachronal wavelengths

We designed and fabricated new samples (5x20 hair array) to study the influence of the wavelengths on the locomotion speed. By curving the soft cilia carpet on magnetization molds with different curvatures (Figure R6a), we can encode different wavelengths into the soft robots. We also consider a sample without a metachronal wave for comparison. These samples are identified by their metachronal wave vector k . A negative k leads to antiplectic waves, and a positive k to symplectic waves. Following the convention, we define the positive direction as the direction of the power stroke. We measure their walking speed in a rotating magnetic field of 40 mT, 30 degrees per second in the x-z plane.

Figure R6: (identical to Figure 8) Influence of metachronal wave vectors on the crawling speed of soft robots. (a) the magnetization moulds that encode antiplectic and symplectic waves on the cilia soft robots. The wave vector and wavelengths are denoted k and λ , while d is equal to 4 mm, which represents the distance between the neighbouring cilia. (b) the waking speeds of soft robots with different metachronal waves. The locomotion speeds are measured based on the geometric centre of the soft robots. (c) the curved substrate on antiplectic and symplectic wave soft robots. The substrate of antiplectic soft robot bulges in the recovery stroke, helping the legs to move forward. On the contrary, the substrate of the symplectic soft robot dents and obstructs the recovery stroke, which slows down the robotic locomotion.

As depicted in Figure R6b, soft robots with antiplectic waves exhibit much higher locomotion speed than the ones with symplectic waves (see also supplementary video 8). We think that this result is due to the different behavior in the bending of the soft substrate (pure Ecoflex) for antiplectic and symplectic waves. Consider the walking gait of a human being (Figure R7); during the stance phase, the leg is pushed backward while being in contact with the ground. During the swing phase, the leg must lift off the ground and move back forward to prepare for the next push backward.

[Redacted]

Figure R7: illustration of the walking gait of human. (image source: <http://www.innovatefpga.com/cgi-bin/innovate/teams.pl?Id=EM040>)

With this analogy in mind, we can understand that for a cilia carpet soft robots to properly locomote, its legs must push back on the ground during the power stroke (equivalent to the stance phase), and lift them up to freely move forward during the recovery stroke (equivalent to the swing phase). Unlike the human walking gait, the distance between the ground and the base of the cilia along the motion for our soft robot depends on the shape of the soft substrate. This shape depends on the torque applied by the cilia, which itself depends on the magnetization pattern of the cilia array, and thus on the wave type. As depicted in Figure R6c, the substrate of the antiplectic wave robot bugles during the recovery stroke, which allow for its legs to efficiently move back forward during the “swing phase”. For symplectic waves however, the substrate dents during the recovery stroke, making it much more difficult for the legs to swing back and thus reducing the efficiency of the walking.

6. The manuscript is lack of a conclusion part.

Thanks for the suggestion, we add the Discussion section (as a conclusion) at the end of the manuscript.

Discussions

We present a soft robotic system of magnetically actuated artificial cilia carpets. Large arrays of artificial cilia can be easily fabricated by curing magnetic composite material in the 3D printed moulds. Both symplectic and antiplectic metachronal waves with various wavelengths can be encoded into the artificial cilia carpets using curved templates in the magnetizer. In this work, the

cilia dimensions are limited by the resolution of the 3d printer. Magnetic actuation of the artificial cilia is limited by the frequency and field strength of the magnetic actuation system. The fluid pumping experiments support previously published numerical results, showing the platform's ability to study complex cilia-fluid interactions at the micrometer-scale. We hope this highly customizable robotic platform can assist studying the collective behaviors of natural cilia and cilia-based active matter systems.

Reviewer #2 (Remarks to the Author):

The paper describes magnetically actuated cilia-like post arrays that exhibit metachronal motion. Whereas similar polymeric posts with embedded magnetic particles have been previously demonstrated, the novelty of the work in differential magnetization of the posts leading to spatially varying magnetic force under a uniform magnetic field. This in turn generate metachronal motion in the post array. The paper demonstrates the applications of such arrays capable of metachronal motion to fluid pumping and robotic locomotion.

We thank the reviewer for appreciating the novelty of our work to encode magnetization into the cilia carpet. Their following comments are very helpful for improving the quality of this work.

1. There is limited information about the properties of the magnetic posts. I was only able to find that they are 4 mm in length in SI. Information about post dimensions and other relevant properties need to be in the main text. Magnetic particle size needs to be also reported.

We thank the reviewer for the reminder to add this key information to the magnetic artificial cilia. We characterize the cilia properties and explain them in the main text as follows:

"The magnetic cilia hair is in a cylinder shape with a diameter of 0.8mm, and length of 4 mm." and "The soft cilia carpets contain non-magnetized NdFeB particles (average diameter 5 μm)." in "Stretchable soft magnetic cilia carpets" section.

We measured the mechanical properties of the magnetic composite material, which composes the body of the cilia. The engineering stress-strain curve for composite materials with different weight ratio composition is shown in Figure R8. It is clear that adding magnetic particles makes the composite material stiffer, as reported in the literature [21]. This magnetic composite material (NdFeB particles + Ecoflex) is very popular for magnetic soft robots [21]-[24], which has been extensively studied.

Figure R8: Mechanical characterization of the Ecoflex-NdFeB composite materials.

2. Comparing to previously published magnetic cilia, the authors claim that the main advantage of their approach is the simplicity of the fabrication method. While this could be arguably true, for practical applications much more important criteria are dimensions of cilia, actuation amplitude, and frequency. These metrics should be included and discussed to position this work with respect to previous publications.

Thank you for the opportunity to clarify our contributions in this paper. Although a few magnetic cilia systems have been reported [25], [26], our system has the following advantages:

1. **Design:** the 3D-printed mold allows us to modify the geometrical parameters (length, width, position) at the individual cilium level.
2. **Modeling:** we provide a predictive model and library of magnetic cilia motions under various parameters
3. **Fabrication:** The materials and molding process are well-established and invariant with hair numbers. This is particularly important to study dense cilia carpet. We have achieved a 100% yield rate on our tested samples. Conventional fabrication methods usually require more time for more hairs, and sometimes requires assembly, making it difficult for a dense cilia array.
4. **Magnetization:** The soft substrate of the cilia carpet allows to encode complex magnetization patterns on the cilia carpet by stretch and fold it on curved surfaces.

More importantly, those advances in design, modeling, fabrication, and magnetization make our artificial cilia system a customizable platform for fundamental cilia research. This level of system integration and complexity was previously only accessible with computer simulations. To prove the capability of this platform, we experimentally confirm two major numerical findings of fluidic transport on cilia carpet for the first time [13]. We believe this platform can facilitate future studies of cilia functions in particle selection, feeding and swimming, and mucus removal in human's respiratory system.

The reviewer raised a very important point here about the hair size. Indeed, compared with previous report artificial cilia systems (length: 10-1000 μm , width: 1-100 μm) and natural cilia (length: 10-100 μm , width: 1 μm), our cilia are very large (length: 4 mm, width: 0.8 mm). Previously reported magnetic cilia systems are developed mostly for microfluidic applications including, mixing and pumping. However, in this work, we claim this soft robotic

system can be useful for fundamental cilia research. For this purpose, the large-size artificial cilia system has a few advantages:

1. The system is easy to fabricate. Microfabrication of micrometer-scale artificial cilia often requires cleanroom facilities. The fabrication process is expensive and takes a long time. On the contrary, 3D printing technologies are cheap and designed for fast prototyping. Most labs can afford a 3D printer.
2. Manipulation. Moving and assembling micrometer objects requires micromanipulators or other microrobots. Here we can directly use our hand to move the carpet if we need to adjust the position.
3. Visualization. Micrometer-scale artificial cilia require a microscope and microparticles to observe cilia motion and track the flow. In this work, we can use standard digital cameras to record both top and side views in the flow. And the fluorescent tracking particles are in diameter of 0.5 μm .

The reason that we can use large-size artificial cilia to study the natural cilia system is that the Reynolds number is very small, so the flow is Stokes flow as in natural cilia systems. We use high viscosity liquid and low-speed motion to ensure a very low Reynolds number ($Re \ll 1$). Considering the size of the cilia carpet, we don't think the method presented in this paper can be directly used for microfluidic applications. Recently, pioneering works show they have successfully made micrometer-scale magnetic cilia using microfabrication for microfluidic applications [27].

As requested by the reviewer, we add figure R12 of the artificial magnetic cilia and compare their properties, and we add this figure to the supplementary materials. (The figure is in the response to your comment #4)

3. Dimensions of the setup for fluid transport experiment are missing. The results are confusing. Previous works showed that velocity above ciliary arrays changes linearly with height. Here, however, velocity decreases to a constant value some distance about cilia (Fig. 4c and Fig. 5d). What is the reason for this constant velocity and why it depends on the wavelength? How was it ensured that the fluorescent particle displacement is not affected by the initial transient and the flow reached steady circulation? It does seem that the experiments were not performed sufficiently accurate.

We thank the reviewer for reminding us of the missing dimensions in the fluidic experiments, we add some key dimensions in Figure 4, and more detailed top and side view of the setup in the supplementary material (Figure R9 reported in the following).

Figure R9, Top and side views of the fluidic setup.

The flow profile of the liquid above the carpet is determined by both the cilia carpet and the boundary conditions of the experimental setup. In this work, we choose this setup design to compare with the numerical studies [13], [28]. This means the cilia carpet is sandwiched between two walls and a thick liquid layer is placed above (about 30 mm). To compare, cilia hairs are 4mm in length). The viscous liquid (99% glycerol) is free to flow in the air-liquid interface. In a similar setup [29], we found the flow profile above the carpet is very similar to our results, as shown in Figure R10 and R11.

[Redacted]

Figure R10, Fluid transport in a similar setup (Figure 8 in [29]).

[Redacted]

Figure R11, Fluid profile in a similar setup with tilted hair samples (Figure 9 in[29]).

As for the second question: "How was it ensured that the fluorescent particle displacement is not affected by the initial transient and the flow reached steady circulation? It does seem that the experiments were not performed sufficiently accurately." We think this is not a problem in our system because the system's Reynolds number is very small ($Re \sim 7e-3$), and the viscosity is so large (glycerol: $1.412 \text{ Pa}\cdot\text{s}$) that the inertia of the fluid is negligible, which means the flow reaches steady condition instantaneously. Each flow experiment is conducted for about 5 minutes, and the cilia beat a full cycle for 12 seconds. During the experiments, we have observed a very consistent performance of the flow within each cycle, as shown in the supplementary video 4, which are typical for the Stokes flow.

4. The velocity of fluid seems to be rather slow. In Fig. 4c and Fig. 5d, it will be more useful to present the velocity data normalized by the length of posts and oscillation period. Furthermore, the results need to be compared to previously published experiments on pumping using artificial cilia.

We thank the reviewer for the constructive suggestion. The comparison of flow speed normalized by the length of posts and oscillation period is a very good idea. We compared with other experimental works in the following figure. Based on the comparison, we can see that our cilia carpets are not particularly slow in normalized pumping speed.

reference	Figure	Cilia length L	Near carpet flow speed v	Frequency f	Normalized speed v/fL
[19]	[Redacted]	30 μm	4 $\mu\text{m/s}$	0.5 Hz	0.27
[29]	[Redacted]	500 μm	500 $\mu\text{m/s}$	7 Hz	0.143
[30]	[Redacted]	350 μm	75 $\mu\text{m/s}$	10 Hz	0.0214
[31]	[Redacted]	25 μm	9 $\mu\text{m/s}$	34 Hz	0.011
This work		4 mm	0.083 mm/s	0.083 Hz	0.25

Figure R12, Comparison of pumping using artificial cilia carpet. The normalized pumping speed v/fL , shows how fast the flow it can generate by one beating cycle.

5. In biological systems, metachronal waves often associated with hydrodynamic coupling between individual cilia. Are there any effects of the fluid on the motion of the magnetic cilia and the wave propagation? How pumping velocity depends on the beating rate? The change of the pumping rate with the frequency need to be explored.

This is a very good question, and reviewer 3 also asked a similar question. In the natural cilia system, the metachronal waves are emergent from cilia local interactions including the hydrodynamic interactions. However, in our system, these metachronal waves are encoded from the magnetization patterns, which is completely deterministic. The fluid has negligible effects on the artificial cilia motion and wave propagation, considering the artificial cilia are beating slowly (one beating cycle needs 12 seconds).

The purpose of the system is to mimic the Stokes' flow conditions to study the fluid dynamics of microscopic cilia, so that moving at low speed does not constitute a problem. If the artificial cilia beat too fast, the Reynolds number might get too high so that turbulent flow occur. In fact, the frequency is limited by the magnetic actuation system (Cardiomag, Multi-Scale Robotics Lab, ETH Zurich) with a maximum frequency of about 1 Hz.

6. The authors claim that metachronal waves are beneficial for fluid pumping. However, there is no discussion of any physical mechanism enhancing fluid pumping to support this. Experiments are also not convincing due to the problems pointed in 3.

We thank the reviewer for raising this important question that has been raised by the first reviewer in their 3rd comment as well. Please check our answers there. I also hope our answers to your previous questions can convince you that the experiments are properly performed.

7. The locomotion is interesting but very briefly investigated. The traveling speed of the structure and its relations to the wavelength and speed, array density need to be reported.

We thank the reviewer for raising this important question. This is another common question that all reviewers asked. We performed new experiments to answer the reviewers' concerns. Please refer to our response to the 5th comment of the reviewer 1. Considering the large design space, we did not experimentally test every possible combination. Optimization of the design parameters is out of the scope of this work.

8. The limitations of the proposed method need to be discussed. It will be especially useful to discuss how these cilia can be miniaturized.

We thank the reviewer for this suggestion, we add the limitations in the discussion section.

“In this work, the cilia dimensions are limited by the resolution of the 3d printer. Magnetic actuation of the artificial cilia is limited by the frequency and field strength of the magnetic actuation system.”

For artificial cilia, there are already some pioneering works of sub-millimeter magnetic artificial cilia using micromolding [26], [27], [32]. We can expect more future works due to their promising applications in microfluidics and microrobotics.

Reviewer #3 (Remarks to the Author):

The manuscript reports a detailed experimental study of the dynamics and transport properties of magnetically actuated, soft, artificial cilia carpets. A new method is presented to fabricate soft, magnetic, cylindrical filaments of sub-millimeter diameter and a few millimeter length, anchored on a deformable membrane substrate. Programmable magnetization patterns can be encoded into these artificial cilia carpets, which then display metachronal wave-like beat patterns in dynamic magnetic fields. The experiments are accompanied by numerical simulations based on elasticity theory. Both transport in a fluid environment and the locomotion (of an "inverted" carpet) on a solid surface have been investigated. The system provides a highly customizable experimental platform. It has been used here to test previous numerical predictions for the efficiency of metachronal transport and the dependence of transport velocity on cilia density. The latter dependence has been difficult to study experimentally so far, due to the difficulty in fabricating samples with various cilia densities.

I think that the current study presents a very significant step forward, both in the fabrication and control of soft robots, and the experimental study of the physics of metachronal dynamics and transport.

We would like to express our sincere appreciation to the reviewer's comments, especially highlighting the fabrication and control of the soft robots and the potential as a platform for future cilia research.

Therefore, I support the publication of the manuscript in Nat. Comm., after the authors have considered the following questions and comments:
(1) What is the length of the cilia?

4 mm in length and 0.8 mm in diameter. Now we add them in the beginning of the results section. We also add more description of the magnetic cilia as other reviewers suggested.

(2) There is one big difference between metachronal waves in artificial and biological systems, which is that these waves are actuated in the former, but self-organized in the latter system. This has apparently (and not unexpectedly) little effect on the transport properties. Nevertheless, it would be good to point out this difference clearly.

This is very good suggestion, the metachronal waves in our system is completely deterministic. It is not self-organized. We clarify it in the text as following:

"Unlike self-organized metachronal waves in natural cilia, the metachronal waves the magnetic cilia carpets are only determined by their magnetization patterns."

(3) Why does the longer power-stroke phase make the metachronal waves antiplectic?

This is a great question. After further investigation, we realized that the wave type (antiplectic/symplectic) is in fact not related to the length of the power-stroke. The metachronal wave types (or more general wave vectors) are actually related to the local curvature of magnetization mold. The curvature along the carpet direction is exactly the wave vector $\kappa = \frac{1}{R} = \frac{1}{L/2\pi} = \frac{2\pi}{L} = k$. By inverting the local curvature, as depicted in figure R13, we can encode symplectic metachronal waves as well on the 8x8 artificial cilia carpets. The longer power-stroke phase this only influences the waveform within a period.

Thank you so much for this question, which inspired us to find methods to encode symplectic waves on the artificial cilia carpets. By expanding the methods to encode symplectic and antiplectic waves on the artificial cilia carpets, we think it has significantly improved the quality of this work.

Figure R13, Comparison between symplectic and antiplectic waves on the 8x8 magnetic artificial cilia carpets. **a**, Encoded symplectic and antiplectic waves on the cilia carpets. The magnetization directions of individual hair are illustrated by the purple arrow. **b**, Snapshots of the 8x8 cilia carpet under one period of rotating magnetic field of 80 mT at 30 degrees per second clockwise in the x-z plane. The red dots represent the straight hair (magnetization direction matches the external magnetic field) in the current frame, as a marker to identify the wave traveling direction.

(4) I find the notation zero wavelength, quarter wavelength, etc. in Fig. 4 confusing. I guess this means infinite wavelength, wavelength of 1/4 of systems size, etc. Please clarify.

Thank you for pointing out this mistake. We correct this notation based on your suggestion. Now we marked them as $\lambda = \infty$, $\lambda = 4L$, $\lambda = 2L$, and $\lambda = L$.

(5) The section of "soft robotic locomotion" is very short and difficult to understand.

We thank the reviewer for this suggestion, we have expanded the "soft robotic locomotion" section and systematically study the relationship locomotion with the metachronal wavelengths.

-- What is the potential for biomimetic applications inside the HUMAN body? The references give no indications of such applications. Millimeter-size cilia seem to be quite large for such applications.

Soft robots at the millimeter-to-centimeter scale have been studied recently for biomedical applications inside the human body[33], [34]. Unlike previously reported microrobots that are mostly versioned to swim inside the blood vessels, the applications of millimeter soft robots can be used in the gastrointestinal tract. Because the body is soft, those soft robots can be folded and wrapped inside a capsule and swallowed to enter the body, and then

perform navigation [35], re-orient and inject drug inside the stomach [36], gripping batteries inside the stomach [37], and sampling tissues [38]. You are right that our walking millipede-inspired soft robot may be too large for such applications, but the fabrication and modeling cover the range of millimeter to centimeter scale soft robots, which can be useful for future designs for functional medical devices that can be magnetically controlled.

-- There is a rolling mechanism indicated at the top of Fig. 6a, which consists of a periodic curling and opening-up time sequence. What is the characteristic frequency? How does it depend on the magnetic field parameters? Why does the robot not simply stay permanently in the rolled-up state?

This is a very good question. Regarding the crawling to rolling transition, there are three types of forces and torques involved in this system: gravity, elastic bending torque from the soft carpet, and magnetic torques and forces. The reason for this repeated curling and opening-up is due to the fact that gravity is too large for the robot to hold up the curved body at certain positions. The frequency of this curling-and-opening motion is determined by the rotating magnetic field.

To show this transition, we use a simplified 2D to show this transition of the soft cilia carpet with encoded magnetization patterns. In the simulation, we confined the carpet in a two-dimensional plane with one fixed end (position and angle). We then simulate 360-degree-counterclockwise rotating magnetic field with different magnitude, under quasi-static conditions. As depicted in Figure R14, the carpet will not rotate if the magnetic field is too weak (20 and 40 mT), and the carpet will roll up if the magnetic field is strong enough to overcome gravity and create a roll (superior or equal to 50 mT in this case).

Figure R14, Simulations of the transition of rolling and crawling. The external magnetic field is represented by the red line. Magnetic torque, elastic modulus, and gravity are considered in the model. The soft carpet is 36 mm in length and 1 mm in width.

-- The authors state that "... locomotion ... does not solely rely on the individual legs, but instead on the wave pattern." This is surprising and interesting, since the waveform (symplectic, antiplletic, ...) makes little difference for fluid transport of cilia carpets. Do soft robots move in opposite directions for symplectic and antiplletic waves?

We thank the reviewer for this comment. First, we would like to clarify that we don't mean that the metachronal wave pattern will directly decide the walking speed. We just want to highlight the redundancy of the system, i.e. that not all individual legs are required to move precisely for the locomotion. Inspired by all three reviewers' comments, we investigate the

influence of metachronal wavelength on the walking speed. The results are shown in the response to comment #5 from the first reviewer.

References:

- [1] J. Edelmann, A. J. Petruska, and B. J. Nelson, "Magnetic control of continuum devices," *Int. J. Rob. Res.*, vol. 36, no. 1, pp. 68–85, Jan. 2017, doi: 10.1177/0278364916683443.
- [2] J. Blake, "A model for the micro-structure in ciliated organisms," *J. Fluid Mech.*, vol. 55, no. 1, pp. 1–23, 1972, doi: 10.1017/S0022112072001612.
- [3] Y. W. Kim and R. R. Netz, "Pumping fluids with periodically beating grafted elastic filaments," *Phys. Rev. Lett.*, vol. 96, no. 15, p. 158101, Apr. 2006, doi: 10.1103/PhysRevLett.96.158101.
- [4] H. Guo, J. Nawroth, Y. Ding, and E. Kanso, "Cilia beating patterns are not hydrodynamically optimal," *Phys. Fluids*, vol. 26, no. 9, p. 091901, Sep. 2014, doi: 10.1063/1.4894855.
- [5] Y. Ding, J. C. Nawroth, M. J. McFall-Ngai, and E. Kanso, "Mixing and transport by ciliary carpets: a numerical study," *J. Fluid Mech.*, vol. 743, no. 2007, pp. 124–140, 2014, doi: 10.1017/jfm.2014.36.
- [6] S. Chateau, J. Favier, U. D'Ortona, and S. Poncet, "Transport efficiency of metachronal waves in 3D cilium arrays immersed in a two-phase flow," *J. Fluid Mech.*, vol. 824, pp. 931–961, Aug. 2017, doi: 10.1017/jfm.2017.352.
- [7] S. Chateau, U. D'Ortona, S. Poncet, and J. Favier, "Transport and Mixing Induced by Beating Cilia in Human Airways," *Front. Physiol.*, vol. 9, no. MAR, p. 161, Mar. 2018, doi: 10.3389/fphys.2018.00161.
- [8] S. Chateau, J. Favier, S. Poncet, and U. D'Ortona, "Why antiplectic metachronal cilia waves are optimal to transport bronchial mucus," *Phys. Rev. E*, 2019, doi: 10.1103/PhysRevE.100.042405.
- [9] S. M. Mitran, "Metachronal wave formation in a model of pulmonary cilia," *Comput. Struct.*, vol. 85, no. 11–14, pp. 763–774, Jun. 2007, doi: 10.1016/j.compstruc.2007.01.015.
- [10] T. Niedermayer, B. Eckhardt, and P. Lenz, "Synchronization, phase locking, and metachronal wave formation in ciliary chains," *Chaos*, vol. 18, no. 3, p. 037128, Sep. 2008, doi: 10.1063/1.2956984.
- [11] E. M. Gauger, M. T. Downton, and H. Stark, "Fluid transport at low Reynolds number with magnetically actuated artificial cilia," *Eur. Phys. J. E*, vol. 28, no. 2, pp. 231–242, Feb. 2009, doi: 10.1140/epje/i2008-10388-1.
- [12] S. N. Khaderi, J. M. J. Den Toonder, and P. R. Onck, "Microfluidic propulsion by the metachronal beating of magnetic artificial cilia: A numerical analysis," *J. Fluid Mech.*, vol. 688, pp. 44–65, 2011, doi: 10.1017/jfm.2011.355.
- [13] N. Osterman and A. Vilfan, "Finding the ciliary beating pattern with optimal efficiency," *Proc. Natl. Acad. Sci.*, vol. 108, no. 38, pp. 15727–15732, 2011, doi:

10.1073/pnas.1107889108.

- [14] P. G. Jayathilake, Z. Tan, D. V. Le, H. P. Lee, and B. C. Khoo, "Three-dimensional numerical simulations of human pulmonary cilia in the periciliary liquid layer by the immersed boundary method," *Comput. Fluids*, vol. 67, pp. 130–137, Aug. 2012, doi: 10.1016/j.compfluid.2012.07.016.
- [15] J. Elgeti and G. Gompper, "Emergence of metachronal waves in cilia arrays," *Proc. Natl. Acad. Sci. U. S. A.*, vol. 110, no. 12, pp. 4470–4475, Mar. 2013, doi: 10.1073/pnas.1218869110.
- [16] J. R. Vélez-Cordero and E. Lauga, "Waving transport and propulsion in a generalized newtonian fluid," *J. Nonnewton. Fluid Mech.*, vol. 199, pp. 37–50, Sep. 2013, doi: 10.1016/j.jnnfm.2013.05.006.
- [17] D. R. Brumley, M. Polin, T. J. Pedley, and R. E. Goldstein, "Metachronal waves in the flagellar beating of *Volvox* and their hydrodynamic origin," *J. R. Soc. Interface*, vol. 12, no. 108, Jul. 2015, doi: 10.1098/rsif.2014.1358.
- [18] R. Di Leonardo, A. Búzás, L. Kelemen, G. Vizsnyiczai, L. Oroszi, and P. Ormos, "Hydrodynamic synchronization of light driven microrotors," *Phys. Rev. Lett.*, vol. 109, no. 3, p. 034104, Jul. 2012, doi: 10.1103/PhysRevLett.109.034104.
- [19] M. Vilfan *et al.*, "Self-assembled artificial cilia," *Proc. Natl. Acad. Sci.*, vol. 107, no. 5, pp. 1844–1847, 2010, doi: 10.1073/pnas.0906819106.
- [20] S. L. Tamm, "Ciliary motion in paramecium: A scanning electron microscope study," *J. Cell Biol.*, vol. 55, no. 1, pp. 250–255, Oct. 1972, doi: 10.1083/jcb.55.1.250.
- [21] Y. Kim, H. Yuk, R. Zhao, S. A. Chester, and X. Zhao, "Printing ferromagnetic domains for untethered fast-transforming soft materials," *Nature*, 2018, doi: 10.1038/s41586-018-0185-0.
- [22] V. K. Venkiteswaran, L. F. P. Samaniego, J. Sikorski, and S. Misra, "Bio-inspired terrestrial motion of magnetic soft millirobots," *IEEE Robot. Autom. Lett.*, 2019, doi: 10.1109/LRA.2019.2898040.
- [23] W. Hu, G. Z. Lum, M. Mastrangeli, and M. Sitti, "Small-scale soft-bodied robot with multimodal locomotion," *Nature*, 2018, doi: 10.1038/nature25443.
- [24] Y. Kim, G. A. Parada, S. Liu, and X. Zhao, "Ferromagnetic soft continuum robots," *Sci. Robot.*, 2019, doi: 10.1126/scirobotics.aax7329.
- [25] F. Tsumori, R. Marume, A. Saijou, K. Kudo, T. Osada, and H. Miura, "Metachronal wave of artificial cilia array actuated by applied magnetic field," in *Japanese Journal of Applied Physics*, 2016, vol. 55, no. 6, doi: 10.7567/JJAP.55.06GP19.
- [26] S. Hanasoge, P. J. Hesketh, and A. Alexeev, "Microfluidic pumping using artificial magnetic cilia," *Microsystems Nanoeng.*, vol. 4, no. 1, p. 11, 2018, doi: 10.1038/s41378-018-0010-9.
- [27] S. Zhang, Y. Wang, P. R. Onck, and J. M. J. den Toonder, "Removal of Microparticles by Ciliated Surfaces—an Experimental Study," *Adv. Funct. Mater.*, 2019, doi: 10.1002/adfm.201806434.
- [28] J. Elgeti and G. Gompper, "Emergence of metachronal waves in cilia arrays," *Proc. Natl. Acad. Sci.*, 2013, doi: 10.1073/pnas.1218869110.

- [29] A. Rockenbach and U. Schnakenberg, "The influence of flap inclination angle on fluid transport at ciliated walls," *J. Micromechanics Microengineering*, 2017, doi: 10.1088/0960-1317/27/1/015007.
- [30] S. Zhang, Y. Wang, R. Lavrijsen, P. R. Onck, and J. M. J. den Toonder, "Versatile microfluidic flow generated by moulded magnetic artificial cilia," *Sensors Actuators, B Chem.*, 2018, doi: 10.1016/j.snb.2018.01.189.
- [31] a R. Shields, B. L. Fiser, B. a Evans, M. R. Falvo, S. Washburn, and R. Superfine, "Biomimetic cilia arrays generate simultaneous pumping and mixing regimes.," *Proc. Natl. Acad. Sci. U. S. A.*, vol. 107, no. 36, pp. 15670–15675, 2010, doi: 10.1073/pnas.1005127107.
- [32] S. Hanasoge, P. J. Hesketh, and A. Alexeev, "Metachronal motion of artificial magnetic cilia," *Soft Matter*, vol. 14, no. 19, pp. 3689–3693, 2018, doi: 10.1039/c8sm00549d.
- [33] L. Hines, K. Petersen, G. Z. Lum, and M. Sitti, "Soft Actuators for Small-Scale Robotics," *Advanced Materials*. 2017, doi: 10.1002/adma.201603483.
- [34] M. Sitti, "Miniature soft robots - road to the clinic," *Nature Reviews Materials*. 2018, doi: 10.1038/s41578-018-0001-3.
- [35] S. Miyashita, S. Guitron, S. Li, and D. Rus, "Robotic metamorphosis by origami exoskeletons," *Sci. Robot.*, 2017, doi: 10.1126/scirobotics.aao4369.
- [36] A. Abramson *et al.*, "An ingestible self-orienting system for oral delivery of macromolecules," *Science (80-.)*, 2019, doi: 10.1126/science.aau2277.
- [37] S. Miyashita, S. Guitron, K. Yoshida, S. Li, D. D. Damian, and D. Rus, "Ingestible, controllable, and degradable origami robot for patching stomach wounds," in *Proceedings - IEEE International Conference on Robotics and Automation*, 2016, doi: 10.1109/ICRA.2016.7487222.
- [38] D. Son, H. Gilbert, and M. Sitti, "Magnetically Actuated Soft Capsule Endoscope for Fine-Needle Biopsy," *Soft Robot.*, 2019, doi: 10.1089/soro.2018.0171.

Reviewers' comments:

Reviewer #1 (Remarks to the Author):

The revised manuscript is satisfactory for its publication in NC. Nice work.

Reviewer #2 (Remarks to the Author):

I am not satisfied with the revision regarding the fluid flow experiments. The dimensions of the experimental setup are still missing which makes the experiments unreproducible. The velocity profiles in Figures 4 and 5 seem to contradict the images in 4d and 5c. The images show circulating flow patterns whereas the velocity profiles have a uniform velocity away from the cilia. The velocity profiles should be extended to the free surface above the cilia. The dimensions of the integration area and depth that were used to obtain the fluid velocity are not reported. The thickness of glycerol layer and viscosity are not reported.

The discussion of the method limitations is superficial.

Line 183: zero-wavelength should be infinite wavelength.

Reviewer #3 (Remarks to the Author):

In their resubmittal letter, the authors have addressed all points of my previous report very carefully and in considerable detail. The corresponding changes in the manuscript have significantly improved the -- already very nice -- manuscript. Therefore, I strongly support the publication of the manuscript in its present form.

Reviewer #1 (Remarks to the Author):

The revised manuscript is satisfactory for its publication in NC. Nice work.

Thank you for your recommendation to publish this article.

Reviewer #2 (Remarks to the Author):

I am not satisfied with the revision regarding the fluid flow experiments. The dimensions of the experimental setup are still missing which makes the experiments unreproducible.

As this reviewer requested, we add the dimensions of the fluidic setup in Supplementary Figure 4 (displayed below) and specified them in the Methods.

Supplementary Figure 4, Top and side views of the fluidic setup. The cilia carpet is placed in a 160 mm large, 110 wide and 45 mm deep acrylic box. The observation window is the 160 mm large side of the box. In the middle width of the box (55 mm from the observation window), we place a 90 mm large and 45 mm high white Polypropylene (POM) board. The square cilia carpet (36 mm × 36 mm) is placed between the observation window and the POM board. The box is filled with 99% glycerol (viscosity: 1.15 Pa·s at 20°C) up to a height about 30 mm.

We have also added the dimension of the magnetic actuation system (CardioMag, MSRL) in the Methods. The magnetic actuation system is developed previously in the same lab. It is composed of 8 electromagnets with a calibrated cube workspace (20 cm x 20 cm x 20 cm) in the center. Within this workspace, a uniform magnetic field can be generated in any direction in space with a maximum magnitude of 80 mT.

[Redacted]

Image of CardioMag. This Image is obtained from paper (Petruska AJ, Edelmann J, Nelson BJ. Model-based calibration for magnetic manipulation. IEEE Transactions on Magnetics. 2017 Jan 16;53(7):1-6.)

We have now listed all the dimensions of the fluidic experiments and we hope the request of the reviewer has been answered. In case some more dimensions are needed, we will be happy to provide them.

The velocity profiles in Figures 4 and 5 seem to contradict the images in 4d and 5c. The images show circulating flow patterns whereas the velocity profiles have a uniform velocity away from the cilia. The velocity profiles should be extended to the free surface above the cilia.

We thank the Reviewer for the comment, which allowed us to clarify an important point. First of all, we realized the dimensions of the area (36 mm × 27 mm) in which particle trajectories were represented were missing. This area is 36 mm large and covers all the cilia carpet, and 27 mm high, from the tip of the cilia to the liquid surface. We have now added the dimensions in the caption of Fig. 4 and 5. Consequently, the particle trajectories plotted in Figure 4d and 5c are tracked from the tip of cilia to the liquid surface as in Figure 4b. However, in Figure 4c and 5d, the flow profile is only displayed up to 15 mm above the tip of the cilia carpet. This choice was motivated by the will to emphasize the area where the transport is more affected by the cilia motion, i.e. 4 mm layer on top of the cilia tip.

Second, regarding the contradictory nature of the results, we think a clarification is needed. In Fig. 4c and Fig. 5d we are not plotting the velocity of the particles, but their displacements over a period (12 s) (defined in the Methods section, line 544), averaged over the width of the carpet. Since cilia beating in natural system serves the purpose of promoting fluid transport, we have chosen to quantify the displacement of particles, which we consider a good measure of the cilia capability to move the fluid layer right on top of them. We will now explain why the shape of the profile and the particles trajectories are not contradictory when we consider the particle displacement instead of the velocity. Cilia beating forces the particles to move in a vortex. While the velocity in the lateral ascending/descending part of the vortex would have a positive/negative value, the displacements are only positive and have a symmetric value with respect to the centre of the vortex. For this reason, the displacement profile is flat in the ascending/descending part of the vortex, while

the velocity profile would be zero because the ascending and descending part would compensate. Very close to the tip of the cilia, transport is much faster, resulting a greater and unidirectional displacement. In this region, the velocity and the displacements profiles would have the same shape.

As requested, here we show the full flow profile (displacement over a period 12s) from the tip of the cilia carpet up to the water/air interface. We include it as Supplementary Figure 8.

Supplementary Figure 8, Average displacement over a period from the top of the cilia carpet to the air water interface. The first 15 mm are also reported in Figure 4 (left) and 5 (right).

The dimensions of the integration area and depth that were used to obtain the fluid velocity are not reported.

We thank the reviewer for the comment and clarified in the methods what is the size of the averaging window used. The average displacement profiles are obtained by averaging the displacement maps in the x-direction (the one parallel to the carpet) over the entire length of the carpet (36 mm). The method to calculate the average displacement is explained in

the Method section. The depth of field of our optical system covers the entire fluidic setup (as reported in the schematic of the experimental setup in Fig. 4a) and consequently all the particles moving on top of the cilia carpet are in focus, resulting in displacements maps averaged over the width of the carpet. This information has been added to the Methods.

(line 544) “Particle trajectories analysis

We tracked the trajectories of the fluorescent particles and quantified their displacements over a period (12 s). We constructed maps of displacements by assigning the displacement over a period to the x and y coordinates averaged over the same period (Fig. 4d and 5c) and vertical profiles of displacements by averaging the maps along the x-direction over the entire length of the carpet (36 mm) (Fig. 4c and 5d). To ease the comparison between different geometries, we calculated the average displacement in the 4 mm of fluid above the carpet (Fig 4d, inset and 5d, inset). This average particle displacement within a period is used to evaluate the pumping performance of the cilia carpet. The depth of field of our optical system covers the entire fluidic setup (Fig. 4a). Consequently, all the particles moving on top of the cilia carpet are on focus and displacements maps are averaged over the width of the carpet.”

The thickness of glycerol layer and viscosity are not reported.

In the pumping experiments, the glycerol is filled up at least 30 mm of thickness, (which is about 25 millimeters higher than the tip of the artificial cilia), as in the Supplementary Figure 8.

According to the literature [Segur JB, Oberstar HE. Viscosity of glycerol and its aqueous solutions. *Industrial & Engineering Chemistry*. 1951 Sep;43(9):2117-20.], the viscosity of 99% glycerol is 1150 centipoise (=1.15 Pa·s) at 20 degrees Celsius. Considering the system is under low Reynolds' number environment, and the artificial cilia motion influenced by the viscosity. A slight change in the viscosity will not change the pumping results.

The discussion of the method limitations is superficial.

We have added more details in the discussion section regarding the limitation of the system. Initially, we did not discuss the limitations in depth because they are not unique for this specific system. Instead, they have been reported and known previously to each community (e.g. soft robotics community commonly use 3d printed structures for silicone moulding, so the limitations of the 3d printers are known). Considering the wide backgrounds of readers of Nature communications, we add more details of these limitations in the discussion section:

“In this work, the dimensions of the mould are determined by the resolution of the 3d printer, which sets limitations for the cilia size and the distance between neighboring cilia. Moulding magnetic artificial cilia at micrometer scale has been realized using SU-8 patterned by photolithography. Increasing NdFeB particle concentration in the composite material will

increase the viscosity of the mixture, making it more difficult to process and fill into the 3d printed mould. Magnetic actuation system (CardioMag) is bulky and slow in dynamic responses.”

Line 183: zero-wavelength should be infinite wavelength.

Thank you very much for pointing out this mistake, we corrected it.

Reviewer #3 (Remarks to the Author):

In their resubmittal letter, the authors have addressed all points of my previous report very carefully and in considerable detail. The corresponding changes in the manuscript have significantly the -- already very nice --manuscript. Therefore, I strongly support the publication of the manuscript in its present form.

Thank you for your recommendation and your high evaluation of this article.

REVIEWERS' COMMENTS:

Reviewer #2 (Remarks to the Author):

To quantify pumping performance of cilia, the authors report magnitude of the displacement vectors per oscillation period (averaged over the length of cilia array) for particles seeding the fluid. This quantity is not well situated to characterize pumping, which is the net fluid transport along the cilia array. Consider for a circulatory flow with zero average velocity. The magnitude of the displacement vector will not be zero whereas there is no net fluid transport and therefore no pumping. The authors need to integrate the horizontal component of the displacement vector (not vector magnitude) over the channel height. This will be equivalent to the horizontal flowrate in the channel that should be compared between different cilia layouts and actuation patterns. They can also report how the horizontal component of the displacement vector changes along the vertical direction in the middle of the cilia array. Furthermore, the vectorial displacement field should be presented to provide a more complete picture of the flow field generated by beating cilia.

Reviewer #2 (Remarks to the Author):

To quantify pumping performance of cilia, the authors report magnitude of the displacement vectors per oscillation period (averaged over the length of cilia array) for particles seeding the fluid. This quantity is not well situated to characterize pumping, which is the net fluid transport along the cilia array. Consider for a circulatory flow with zero average velocity. The magnitude of the displacement vector will not be zero whereas there is no net fluid transport and therefore no pumping. The authors need to integrate the horizontal component of the displacement vector (not vector magnitude) over the channel height. This will be equivalent to the horizontal flowrate in the channel that should be compared between different cilia layouts and actuation patterns. They can also report how the horizontal component of the displacement vector changes along the vertical direction in the middle of the cilia array. Furthermore, the vectorial displacement field should be presented to provide a more complete picture of the flow field generated by beating cilia.

We are very grateful for the comments.

We agree with the reviewer that, in the case of a vortex, the overall displacement will not be zero while there is no overall net fluid transport. The displacement allows us to quantify the particle motion on top of the carpet induced by the cilia beating, which in our opinion can be considered as a parameter to evaluate the pumping performance. In some of the experiments (for example 8x8 cilia array with antiplectic wave), a vortex is formed between the water/air interface and the cilia carpet. However, in Figure 4c and 5d, the average displacement in the fluid layer from 0 to 4 mm above the cilia carpet (liquid height is about 30 to 35 mm) allows comparing different geometries or wavelengths of the carpet. Within this range, particles move parallel to the cilia carpet and the vertical component of their displacement is negligible. Thanks to this feature of the flow field on top of the cilia carpet, the average displacement over a beating cycle (12s) can be used to quantify the transport generated by the cilia motion.

In Supplementary Figure 5, we use two examples (8x8 cilia carpet and 4x4 cilia carpet) to compare different quantities to quantify ciliary induced transport: the average displacement over a beating cycle, the magnitude of the displacement vector in the x-direction and its absolute value, as suggested by the Reviewer. 8x8 and 4x4 cilia carpets have the high and the lowest pumping performances, respectively. The blue circles are the magnitude of the displacement vector, the same as in Figure 4c and Figure 5d. Red and yellow circles are the average profile of the displacement vector in the x-direction and its absolute value, respectively. From the figure, we can see that projections of the displacement vectors in x-direction have negative values close to the top and positive values close to the cilia carpet. This plot confirms the presence a vortex on top of the 8x8 cilia carpet, as anticipated by the circular trajectories in Figure 4d and Figure 5c. The difference between the yellow and blue curve is attributable to the y-component of the displacement, which is non-zero on the side of the vortex, where the flow is ascending or descending. However, the small difference confirms that the contribution of the y-component of the displacement is limited. Importantly, the three parameters have, to a good approximation, the same value in the region between [0, 4 mm], where displacement is averaged to compare different cilia carpets. Based on the results presented in this figure, we can assume that the particles move along the x-direction and that using the magnitude of the displacement vector in the x-direction would not change the result presented of Figure 4c inset and 5d inset. However, since we think that the average displacement in the x direction is a valuable information, we have computed their profiles in the same conditions of

Fig. 4 and 5 and added Supplementary Figure 5 and 9 to the Supplementary information. We have also added a comment in the main manuscript:

We also compared different measures of the transport of the tracer particles in Supplementary Figure 5.

Although adding the vectoral displacement fields can provide more intuitive views of the flow generated by the cilia carpets, we think they are not essential for this study. Results presented in Figure 4 and 5 have already confirmed the numerical findings from previous studies, which shows the feasibility of this artificial cilia system to study cilia-liquid interactions at the micrometer scale.

Supplementary Figure 5, Comparison of different measures to quantify the flow in the x-direction. The 8x8 cilia array with antiplectic wave is shown, on the left, as a high-speed flow example. The 4x4 cilia array with antiplectic wave is shown, on the right, as a low-speed flow example. Both cases show that different measures of the transport in the region of [0, 4 mm] are equivalent to a good approximation. Blue circles are the displacement magnitude $\sqrt{\delta x^2 + \delta y^2}$, as shown in Figure 3c and Figure 4d in the manuscript. Red and yellow circles represent the projections of the displacement vector in x-direction, δx , and its absolute value $|\delta x|$, respectively.

Supplementary Figure 9, The averaged projections of displacement vectors in x-direction of the fluorescent particles. (a,b) The tracking region (depicted as the green box) of the tracer particles in the pumping experiments. (c) The displacement in the x-direction over a period of 8x8 cilia carpets with various metachronal wavelengths. The magnitude of the displacement vectors of the same experiments is shown in Figure 4c in the manuscript. (d) The displacement in the x-direction of cilia carpets with different cilia densities over a period. The magnitude of the displacement vectors of the same experiments is shown in Figure 5d in the manuscript. The colored lines in panel c and d are smoothed results of nearest 21 points.